



**Comparative geochemical study on Furongian (Toledanian) and Ordovician**
**(Sardic) felsic magmatic events in south-western Europe**
J. Javier Álvaro[a]*, Teresa Sánchez-García[b], Claudia Puddu[c], Josep Maria Casas[d],
Alejandro Díez-Montes[e], Montserrat Liesa[f] & Giacomo Oggiano[g]
[a] *Instituto de Geociencias (CSIC-UCM), Dr. Severo Ochoa 7, 28040 Madrid, Spain,*
*jj.alvaro@csic.es*
[b] *Instituto Geológico y Minero de España, Ríos Rosas 23, 28003 Madrid, Spain,*
*t.sanchez@igme.es*
[c] *Dpto. Ciencias de la Tierra, Universidad de Zaragoza, 50009 Zaragoza, Spain,*
*claudiapuddugeo@gmail.com*
[d] *Dpt. de Dinàmica de la Terra i de l'Oceà, Universitat de Barcelona, Martí Franquès*
*s/n, 08028 Barcelona, Spain, casas@ub.edu*
[e] *Instituto Geológico y Minero de España, Plaza de la Constitución 1, 37001*
*Salamanca, Spain, al.diez@igme.es*
[f] *Dept. de Mineralogia, Petrologia i Geologia aplicada, Universitat de Barcelona, Martí*
*Franquès s/n, 08028 Barcelona, Spain, mliesa@ub.edu*
[g] *Dipartimento di Scienze della Natura e del Territorio, 07100 Sassari, Italy,*
*giacoggi@uniss.it*
* Corresponding author





**ABSTRACT**

A geochemical comparison of Early Palaeozoic felsic magmatic episodes throughout the south-western European margin of Gondwana is analysed. The comparison is made between (i) Furongian–Early Ordovician (Toledanian) activies recorded in the Central Iberian and Galicia-Trás-os-Montes Zones of the Iberian Massif, and (ii) Early–Late Ordovician (Sardic) activities in the eastern Pyrenees, Occitan Domain (Albigeois, Montagne Noire and Mouthoumet massifs) and Sardinia. Both phases are related to uplift and denudation of an inherited palaeorelief, and stratigraphically preserved as distinct angular discordances and paraconformities involving gaps of up to 30 m.y. The geochemical features of the Toledanian and Sardic, felsic-dominant activies point to a predominance of byproducts derived from the melting of metasedimentary rocks, rich in $SiO_2$ and $K_2O$ and with peraluminous character. $Zr/TiO_2$, $Zr/Nb$, $Nb/Y$ and $Zr$ vs. $Ga/Al$ ratios, and REE and $\varepsilon Nd$ values suggest the contemporaneity, for both phases, of two geochemical scenarios characterized by arc and extensional features evolving to distinct extensional and rifting conditions associated with the final outpouring of mafic tholeiitic-dominant lava flows. The Toledanian and Sardic phases are linked to neither metamorphism nor penetrative deformation; on the contrary, their unconformities are associated with foliation-free open folds subsequently affected by the Variscan deformation. The geochemical and structural framework precludes a subduction scenario reaching the crust in a magmatic arc to back-arc setting, but favours partial melting of sediments and/or granitoids in a continental lower crust triggered by the underplating of hot mafic magmas during extensional events related to the opening of the Rheic Ocean.

**Keywords**: granite, orthogneiss, geochemistry, Cambrian, Ordovician, Gondwana.





## 1. Introduction

A succession of stepwise Early–Palaeozoic magmatic episodes, ranging in age from Furongian to Late Ordovician, is widespread along the south-western European margin of Gondwana. Magmatic pulses are characterized by their preferential development in different palaeogeographic areas and linked to the development of stratigraphic unconformities, but they are related to neither metamorphism nor penetrative deformation. In the Central Iberian Zone of the Iberian Massif (representing the western branch of the Ibero-Armorican Arc), this magmatism is mainly represented by the Ollo de Sapo Formation, which has long been recognized as a Furongian–Early Ordovician (495–470 Ma) assemblage of felsic-dominant volcanic, subvolcanic and plutonic igneous rocks. This magmatic activity is contemporaneous with the development of the Toledanian Phase, which places Lower Ordovician (upper Tremadocian–Floian) rocks onlapping an inherited palaeorelief formed by Ediacaran–Cambrian rocks and involving a sedimentary gap of ca. 22 m.y. This unconformity can be correlated with the "Furongian gap" identified in the Ossa-Morena Zone of the Iberian Massif and in the Anti-Atlas Ranges of Morocco (Álvaro et al., 2007, 2018; Álvaro and Vizcaïno, 2018; Sánchez-García et al., 2019) and with the "lacaune normande" in the central and North-Armorican Domains (Le Corre et al., 1991).

Another felsic-dominant magmatic event, although younger (Early–Late Ordovician) in age, has been recognized in some massifs situated along the eastern branch of the Variscan Ibero-Armorican Arc, such as the eastern Pyrenees, the Occitan Domain and Sardinia. This magmatism is related to the Sardic unconformity, where Furongian–Lower Ordovician rocks are unconformably overlain by those attributed to the Sandbian–lower Katian (former Caradoc). The Sardic Phase is related to a sedimentary gap of ca. 16–20 m.y. and geometrically ranges from 90° (angular discordance) to 0° (paraconformity) (Barca and Cherchi, 2004; Funneda and Oggiano, 2009; Álvaro et al., 2016, 2018; Casas et al., 2019).

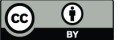



Although a general consensus exists to associate this Furongian–Ordovician
magmatism with the opening of the Rheic Ocean and the drift of Avalonia from
northwestern Gondwana (Díez Montes et al*.,* 2010; Nance et al., 2010; Thomson et al*.,*
2010; Álvaro et al., 2014a), the origin of this magmatism has received different
interpretations. In the Central Iberian Zone, for instance, proposals point to: (i) magmas
formed in a subduction scenario reaching the crust in a magmatic arc to back-arc
setting (Valverde-Vaquero and Dunning, 2000; Castro et al*.,* 2009); (ii) magmas
resulting from partial melting of sediments or granitoids in a continental lower crust
affected by the underplating of hot mafic magmas during an extensional regime (Bea et
al., 2007; Montero et al., 2009; Díez Montes et al., 2010); and (iii) magmas formed by
post-collisional decompression melting of an earlier thickened continental crust, and
without significant mantellic involvement (Villaseca et al., 2016). In the Occitan Domain
(southern French Massif Central and Mouthoumet massifs) and the eastern Pyrenees,
Marini (1988), Pouclet et al. (2017) and Puddu et al. (2019) have suggested a link to
mantle thermal anomalies. Navidad et al. (2018) proposed that the Pyrenean
magmatism was induced by progressive crustal thinning and uplift of lithospheric
mantle isoterms. In Sardinia, Oggiano et al. (2010), Carmignani et al. (2001), Gaggero
et al. (2012) and Cruciani et al. (2018) have suggested that a subduction scenario,
mirroring an Andean-type active margin, originated the main Mid–Ordovician magmatic
activity. In the Alps, the Sardic counterpart is also interpreted as a result of the collision
of the so-called Qaidam Arc with this Gondwanan margin, subsequently followed by the
accretion of the Qilian Block (Von Raumer and Stampfli, 2008; Von Raumer et al.,
2013, 2015). This geodynamic interpretation is mainly suggested for the Alpine
Briançonnais-Austroalpine basement, where the volcanosedimentary complexes
postdating the Sardic tectonic inversion and folding stage portray a younger arc-arc
oblique collision (450 Ma) of the eastern tail of the internal Alpine margin with the Hun
terrane, succeeded by conspicuous exhumation in a transform margin setting (430 Ma)





(Zurbriggen et al., 1997; Schaltegger et al., 2003; Franz and Romer, 2007; Von
Raumer and Stampfli, 2008; Von Raumer et al., 2013; Zurbriggen, 2015, 2017).
Till now the Toledanian and Sardic magmatism had been studied and interpreted
separately on different areas without taking into account their similarities and
differences. In this work, the geochemical affinities of the Furongian–Early Ordovician
(Toledanian) and Early–Late Ordovician (Sardic) felsic magmatic activities recorded in
the Central Iberian and Galicia-Trás-os-Montes Zones, Pyrenees, Occitan Domain and
Sardinia are compared. This re-appraisal may contribute to a better understanding of
the meaning and origin of this stepwise magmatism, and thus, to discuss the
geodynamic scenario of this Gondwana margin (Fig. 1A) during Cambrian–Ordovician
times, bracketed between the Cadomian and Variscan orogenies.

**2. Geological setting of magmatic events**

The following description follows a SW-NE palaeogeographic transect throughout the
south-western European margin of Gondwana during Cambro–Ordovician times.

**2.1. Central Iberian and Galicia-Trás-os-Montes Zones**

In the Ossa Morena and southern Central Iberian Zones of the Iberian Massif (Fig. 1A–
B), the so-called Toledanian Phase is recognized as an angular discordance that
separates variably tilted Ediacaran–Cambrian Series 2 rifting volcanosedimentary
packages from overlying passive-margin successions. The Toledanian gap comprises,
at least, most of the Furongian and basal Ordovician, but the involved erosion can
incise into the entire Cambrian and the upper Ediacaran Cadomian basement
(Gutiérrez-Marco et al., 2019; Álvaro et al., 2019; Sánchez-García et al., 2019).
Recently, Sánchez-García et al. (2019) have interpreted the Toledanian Phase as a
break-up (or rift/drift) unconformity with the Armorican Quartzite (including the Purple



134 Series and Los Montes Beds; McDougall et al., 1987; Gutiérrez-Alonso et al., 2007;

135 Shaw et al., 2012, 2014) sealing an inherited Toledanian palaeorelief (Fig. 2).

136  The phase of uplift and denudation of an inherited palaeorelief composed of upper

137 Ediacaran–Cambrian rocks is associated with the massive outpouring of felsic-

138 dominant calc-alkaline magmatic episodes related to neither metamorphic nor cleavage

139 features. This magmatic activity is widely distributed throughout several areas of the

140 Iberian Massif, such as the Cantabrian Zone and the easternmost flank of the West

141 Asturian-Leonese Zone, where sills and rhyolitic lava flows and volcaniclastics mark

142 the base of the Armorican Quartzite (dated at 477.5 ± 0.9 Ma; Gutiérrez-Alonso et al.,

143 2007, 2016), and the lower Tremadocian Borrachón Formation of the Iberian Chains

144 (Álvaro et al., 2008). Similar ages have been reported in the igneous rocks of the Basal

145 Allochthonous Units and the Schistose Domain in the Galicia-Trás-os-Montes Zone

146 (500–462 Ma; Valverde-Vaquero et al., 2005, 2007; Montero et al., 2009; Talavera et

147 al., 2008, 2013; Dias da Silva et al., 2012, 2014; Díez Fernández et al., 2012; Farias et

148 al., 2014) and different areas of the Central Iberian Zone, including the contact

149 between the Central Iberian and Ossa-Morena Zones, where the Carrascal and

150 Portalegre batoliths are intruded and the felsic volcanosedimentary Urra Formation is

151 interbedded (494–470 Ma, Solá et al., 2008; Antunes et al., 2009; Neiva et al., 2009;

152 Romaõ et al., 2010; Rubio-Ordóñez et al., 2012; Villaseca et al., 2013) (Fig. 1B).

153  The most voluminous Toledanian-related volcanic episode is represented by the

154 Ollo de Sapo Formation, which covers the northeastern Central Iberian Zone. It mainly

155 consists of felsic volcanosedimentary and volcanic rocks interbedded at the base of the

156 Lower Ordovician strata and plutonic bodies. The Ollo de Sapo volcanosedimentary

157 Formation has long been recognized as an enigmatic Furongian–Early Ordovician

158 (495–470 Ma) magmatic event exposed along the core of a 600 km-long antiform

159 (labelled as *77* in Fig. 1B) (Valverde-Vaquero and Dunning, 2000; Bea et al., 2006;

160 Montero et al., 2007, 2009; Zeck et al., 2007; Castiñeiras et al., 2008a; Díez Montes et

161 al., 2010; Navidad and Castiñeiras, 2011; Talavera et al., 2013; López-Sánchez et al.,





2015; Díaz-Alvarado el al., 2016; Villaseca et al., 2016; García-Arias et al., 2018). The
peak of magmatic activity was reached at ca. 490–485 Ma and its most recognizable
characteristic is the presence of abundant megacrysts of K-feldspar, plagioclase and
blue quartz. There is no evident space-time relationship in its distribution (for a
discussion, see López-Sánchez et al., 2015) and, collectively, the Ollo de Sapo
Formation rocks constitute a major tectonothermal event whose expression can be
found in most of the Variscan massifs of continental Europe including the Armorican
and Bohemian massifs (e.g., von Quadt, 1997; Kröner and Willmer, 1998; Linnemann
et al., 2000; Tichomirowa et al., 2001; Friedl et al., 2004; Mingram et al., 2004; Teipel
et al., 2004; Ballèvre et al., 2012; El Korh et al., 2012; Tichomirowa et al., 2012; for a
summary, see Casas and Murphy, 2018). The large amount of magmatic rocks located
in the European Variscan Belt led some authors to propose the existence of a siliceous
Large Igneous Province (LIP) (Díez Montes et al., 2010; Gutiérrez-Alonso et al., 2016),
named Ibero-Armorican LIP by García-Arias et al. (2018).

**2.2. Eastern Pyrenees**

In the eastern Pyrenees, earliest Ordovician volcanic-free passive-margin conditions,
represented by the Jujols Group (Padel et al., 2018), were followed by a late Early–Mid
Ordovician phase of uplift and erosion that led to the onset of the Sardic unconformity
(Fig. 2). Uplift was associated with magmatic activity, which pursuit until Late
Ordovician times. An extensional pulsation took place then developing normal faults
that controlled the sedimentation of post–Sardic siliciclastic deposits infilling
palaeorelief depressions. Acritarchs recovered in the uppermost part of the Jujols
Group suggest a broad Furongian–earliest Ordovician age (Casas and Palacios, 2012),
conterminous with a maximum depositional age of ca. 475 Ma, based on the age of the
youngest detrital zircon populations (Margalef et al., 2016). On the other hand, a ca.
459 Ma U–Pb age for the Upper Ordovician volcanic rocks overlying the Sardic





Unconformity has been proposed in the eastern Pyrenees (Martí et al., 2019), and ca.
455 and 452 Ma in the neighbouring Catalan Coastal Ranges, which represents the
southern prolongation of the Pyrenees (Navidad et al., 2010; Martínez et al., 2011).
Thus, a time gap of about 16–23 m.y. can be related to the Sardic Phase in the eastern
Pyrenees and the neighbouring Catalan Coastal Ranges.

Coeval with the late Early–Mid Ordovician phase of generalized uplift and

denudation, a key magmatic activity led to the intrusion of voluminous granitoids, about
500 to 3000 m thick and encased in strata of the Ediacaran–Lower Ordovician
Canaveilles and Jujols groups (Fig. 2). These granitoids constitute the protoliths of the
large orthogneissic laccoliths that punctuate the backbone of the eastern Pyrenees.
These are, from west to east (Fig. 1D), the Aston (470 ± 6 Ma, Denèle et al., 2009; 467
± 2 Ma, Mezger and Gerdes, 2016), Hospitalet (472 ± 2 Ma, Denèle et al., 2009),
Canigó (472 ± 6 Ma, Cocherie et al., 2005; 462 ± 1.6 Ma, Navidad et al., 2018), Roc de
Frausa (477 ± 4 Ma, Cocherie et al., 2005; 476 ± 5 Ma, Castiñeiras et al., 2008b) and
Albera (470 ± 3 Ma, Liesa et al., 2011) massifs, which comprise a dominant Floian–
Dapingian age. It is noticeable the fact that only a minor representation of coeval basic
magmatic rocks are outcropped. The acidic volcanic equivalents have been
documented in the Albera massif, where subvolcanic rhyolitic porphyroid rocks have
yielded similar ages to those of the main gneissic bodies at 465 ± 4, 472 ± 3, 473 ± 2
and 474 ± 3 Ma (Liesa et al., 2011). Similar acidic byproducts are represented by the
rhyolitic sills of Pierrefite (Calvet et al., 1988).

The late Early–Mid Ordovician ("Sardic") phase of uplift was succeeded by a Late

Ordovician extensional pulsation responsible for the opening of (half-)grabens infilled
with the basal Upper Ordovician alluvial-to-fluvial conglomerates (La Rabassa
Formation). At cartographic scale, a set of NE-SW trending normal faults abruptly
disturbing the thickness of the basal Upper Ordovician formations can be recognized in
the La Cerdanya area (Casas and Fernández, 2007; Casas, 2010). Sharp variations in
the thickness of the Upper Ordovician strata have been documented by Hartevelt



(1970) and Casas and Fernández (2007). Drastic variations in grain size and thickness
can be attributed to the development of palaeotopographies controlled by faults and
subsequent erosion of uplifted palaeoreliefs, with subsequent infill of depressed areas
by alluvial fan and fluvial deposition, finally sealed by Silurian sediments (Puddu et al.,
2019). A Late Ordovician magmatic pulsation contemporaneously yielded a varied set
of magmatic rocks. Small granitic bodies are encased in the Canaveilles and Jujols
strata of the Canigó massif. They constitute the protoliths of the Cadí (456 ± 5 Ma,
Casas et al., 2010), Casemí (446 ± 5 and 452 ± 5 Ma, Casas et al., 2010), Núria (457 ±
4 and 457 ± 5 Ma, Martínez et al., 2011) and Canigó G-1 type (457 ± 1.6 Ma, Navidad
et al., 2018) gneisses.
The lowermost part of the Canaveilles Group (the so-called Balaig Series) host
metre-scale thick bodies of metadiorite interbeds related to an Upper Ordovician
protolith, (453 ± 4 Ma, SHRIMP U–Pb in zircon, Casas et al., 2010). Coeval calc-
alkaline ignimbrites, andesites and volcaniclastic rocks are interbedded in the Upper
Ordovician succession of the Bruguera and Ribes de Freser areas (Robert and
Thiebaut, 1976; Ayora, 1980; Robert, 1980; Martí et al., 1986, 2019). In the Ribes area,
a granitic body with granophyric texture, dated at 458 ± 3 Ma by Martínez et al. (2011),
intruded at the base of the Upper Ordovician succession. In the La Pallaresa dome,
some metre-scale rhyodacitic to dacitic subvolcanic sills, Late Ordovician in age (ca.
453 Ma, Clariana et al., 2018), occur interbedded within the pre-unconformity strata
and close to the base of the Upper Ordovician.

**2.3. Occitan Domain: Albigeois, Montagne Noire and Mouthoumet massifs**

The parautochthonous framework of the southern French Massif Central, named
Occitan Domain by Pouclet et al., (2017), includes among others, from south to north,
the Mouthoumet, Montagne Noire and Albigeois massifs. The domain represents an
eastern prolongation of the Variscan South Armorican Zone (including southwestern





Bretagne and Vendée). Since Gèze (1949) and Arthaud (1970), the southern edge of
the French Massif Central has been traditionally subdivided, from north to south, into
the northern, axial and southern Montagne Noire (Fig. 1C). The Palaeozoic succession
of the northern and southern sides includes sediments ranging from late Ediacaran to
Silurian and from Terreneuvian (Cambrian) to Visean in age, respectively. These
successions are affected by large scale, south-verging recumbent folds that display a
low to moderate metamorphic grade. Their emplacement took place in Late Visean to
Namurian times (Engel et al., 1980; Feist and Galtier, 1985; Echtler and Malavieille,
1990). The Axial Zone consists of plutonic, migmatitic and metamorphic rocks globally
arranged in a bulk dome oriented ENE-WSW (Fig. 1C), where four principal lithological
units can be recognized (i) schists and micaschists, (ii) migmatitic orthogneisses, (iii)
metapelitic metatexites, and (iv) diatexites and granites (Cocherie, 2003; Faure et al.,
2004; Roger et al., 2004, 2015; Bé Mézème, 2005; Charles et al., 2009; Rabin et al.,
2015). The Rosis micaschist synform subdivides the eastern Axial Zone into the
Espinouse and Caroux sub-domes, whereas the southwestern edge of the Axial Zone
comprises the Nore massif.
In the Occitan Domain, two main Cambro–Ordovician felsic events can be identified
giving rise to the protoliths of (i) the Larroque metarhyolites in the northern Montagne
Noire and Albigeois, thrusted from Rouergue; and (ii) the migmatitic ortogneisses in the
Axial Zone of the Montagne Noire (Fig. 2).
(i) The Larroque volcanosedimentary Complex is a thick (500–1000 m) package of
porphyroclastic metarhyolites located on the northern Montagne Noire (Lacaune
Mountains), Albigeois (St-Salvi-de-Carcavès and St-Sernin-sur-Rance nappes) and
Rouergue; the Variscan setting of the formation is allochthonous in the Albigeois and
parautochthonous in the rest. This volcanism emplaced above the Furongian strata and
the so-called "Série schisto-gréseuse verte" (see Guérangé-Lozes et al., 1996;
Guérangé-Lozes and Alabouvette, 1999), and is encased in the upper part of the
Miaolingian La Gardie Formation (Pouclet et al., 2017) (Fig. 2). The Larroque volcanic



rocks consist of deformed microgranites or porphyroclastic rhyolites rich in largely
fragmented, lacunous (rhyolitic) quartz and alkali feldspar phenocrysts. The
metarhyolites occur as porphyritic lava flows, sills and other associated facies, such as
aphyric lava flows, porphyritic and aphyric pyroclastic flows of welded or unwelded
ignimbritic types, fine to coarse tephra deposits, and epiclastic and volcaniclastic
deposits. Although these rocks are also named "augen gneiss", they do not display a
high-grade gneiss paragenesis but a general lower grade metamorphic mineralogy.
The Occitan augen gneisses mimic the "Ollo de Sapo" facies from the Central Iberian
Zone because of their large bluish quartz phenocrysts. Based on geochemical
similarities and contemporaneous emplacement, Pouclet et al. (2017) suggested that
this event also supplied the Davejean acidic volcanic rocks in the Mouthoumet Massif,
which represent the southern prolongation of the southern Montagne Noire (Fig. 2),
and the Génis rhyolitic unit of the western Limousin sector.
(ii) Some migmatitic orthogneisses make up the southern Axial Zone, from the
western Cabardès to the eastern Caroux domes. The orthogneisses, derived from
Ordovician metagranites bearing large K-feldspar phenocrysts, were emplaced at 471
± 4 Ma (Somail Orthogneiss, Cocherie et al., 2005), 456 ± 3 and 450 ± 6 Ma (Pont de
Larn and Gorges d'Héric gneisses, Roger et al., 2004) and 455 ± 2 Ma (Sain Eutrope
gneiss, Pitra et al., 2012). They intruded a metasedimentary pile, traditionally known as
"Schistes X" and formally named St. Pons-Cabardès Group (Fig. 2). The latter consists
of schists, greywackes, quartzites and subsidiary volcanic tuffs and marbles (Demange
et al., 1996; Demange, 1999; Alabouvette et al., 2003; Roger et al., 2004; Cocherie et
al., 2005). The group is topped by the Sériès Tuff, dated at 545 ± 15 Ma (Lescuyer and
Cocherie, 1992), which represents a contemporaneous equivalent of the Cadomian
Rivernous rhyolitic tuff (542.5 ± 1 and 537.1 ± 2.5 Ma) from the Lodève inlier of the
northern Montagne Noire (Álvaro et al., 2014b, 2018; Padel et al., 2017). Migmatization
has been dated by monacites from migmatites and anatectic granites at 327 ± 7, 333 ±



6 and 333 ± 4 Ma (Bé Mézème, 2005; Charles et al., 2008); as a result, the 330–325
Ma time interval can represent a Variscan crustal melting event in the Axial Zone.
As in the Pyrenees, the Middle Ordovician is absent in the Occitan Domain. Its gap
allows distinction between a Lower Ordovician pre-unconformity sedimentary package
para- to unconformably overlain by an Upper Ordovician–Silurian succession (Álvaro et
al., 2016; Pouclet et al., 2017).

**2.4. Sardinia**

In Sardinia the Cambro–Ordovician magmatism is well represented in the external
(southern) and internal (northern) nappe zones of the exposed Variscan Belt (Fig. 1E),
and ranges in age from late Furongian to Late Ordovician. A Furongian–Tremadocian
(ca. 491–480 Ma) magmatic activity, predating the Sardic phase, is mostly represented
by felsic volcanic and subvolcanic rocks encased in the San Vito sandstone Formation.
The Sardic-related volcanic products differ from one nappe to another: intermediate
and basic (mostly metandesites and andesitic basalts) are common in the nappe
stacking of the central part of the island (Barbagia and Goceano), whereas felsic
metavolcanites prevail in the southeastern units. Their age is bracketed between 465
and 455 Ma (Giacomini et al., 2006; Oggiano et al., 2010; Pavanetto et al., 2012;
Cruciani et al., 2018) and matches the Sardic gap based on biostratigraphy (Barca et
al., 1988).
Teichmüller (1931) and Stille (1939) were the first to recognize in southwestern
Sardinia an intra–Ordovician stratigraphic hiatus. Its linked erosive unconformity is
supported by a correlatable strong angular discordance in the Palaeozoic basement of
the Iglesiente-Sulcis area, External Zone (Carmignani et al., 2001). This major
discontinuity separates the Cambrian–Lower Ordovician Nebida, Gonnesa and Iglesias
groups (Pillola et al., 1998) from the overlying coarse-grained ("Puddinga") Monte
Argentu metasediments (Leone et al., 1991, 2002; Laske et al., 1994). The gap





comprises a chronostratigraphically constrained minimum gap of about 18 m.y. that
includes the Floian and Dapingian (Barca et al., 1987, 1988; Pillola et al., 1998; Barca
and Cherchi, 2004) (Fig. 2). The hiatus is related to neither metamorphism nor
cleavage, though some E-W folds have been documented in the Gonnesa Anticline
and the Iglesias Syncline (Cocco et al., 2018), which are overstepped by the
"Puddinga" metaconglomerates. Both the E-W folds and the overlying
metaconglomerates were subsequently affected by Variscan N-S folds (Cocco and
Funneda, 2011, 2017). Sardic-related volcanic rocks are not involved in this area, but
Sardic-inherited palaeoreliefs are lined with breccia slides that include metre- to
decametre-scale carbonate boulders ("Olistoliti"), some of them hosting
synsedimentary faults contemporaneously mineralized with ore bodies (Boni and
Koeppel, 1985; Boni, 1986; Barca, 1991; Caron et al., 1997). The lower part of the
unconformably overlying Monte Argentu Formation deposited in alluvial to fluvial
environments (Martini et al., 1991; Loi et al., 1992; Loi and Dabard, 1997).
A similar gap was reported by Calvino (1972) in the Sarrabus-Gerrei units of the
External Nappe Zone. The so-called "Sarrabese Phase" is related to the onset of thick,
up to 500 m thick, volcanosedimentary complexes and volcanites (Barca et al., 1986;
Di Pisa et al., 1992) with a Darriwilian age for the protoliths of the metavolcanic rocks
(464 ± 1 Ma, Giacomini et al., 2006; 465.4 ± 1.9 Ma, Oggiano et al., 2010). In the
Iglesiente-Sulcis region (Fig. 1E), Carmignani et al. (1986, 1992, 1994, 2001)
suggested that the "Sardic-Sarrabese phase" should be associated with the
compression of a Cambro–Ordovician back-arc basin that originated the migration of
the Ordovician volcanic arc toward the Gondwanan margin.
Some gneissic bodies, interpreted as the plutonic counterpart of metavolcanic rocks,
are located in the Bithia unit (Monte Filau areas, 457.5 ± 0,33 and 458.21 ± 0.32 Ma,
Pavanetto et al., 2012) and in the internal units (Lodè orthogneiss, 456 ± 14 Ma;
Tanaunella orthogneiss, 458 ± 7 Ma, Helbing and Tiepolo, 2005; Golfo Aranci
orthogneiss, 469 ± 3.7 Ma, Giacomini et al., 2006).





The Sardic palaeorelief is sealed by Upper Ordovician trangressive deposits. The
sedimentary facies show high variability, but the –mostly terrigenous– sediments vary
from grey fine- to medium-sized sandstones, to muddy sandstones and mudstones.
They are referred to the Katian Punta Serpeddì and Orroeledu formations (Pistis et al.,
2016). This post–Sardic sedimentary succession is coeval with a new magmatic
pulsation represented by alkaline to tholeiitic within-plate basalts (Di Pisa et al., 1992;
Gaggero et al., 2012).

**3. Geochemical data**

The rocks selected for geochemical analysis (231 samples; geographically settled in
Fig. 1 and stratigraphically in Fig. 2) have recorded different degrees of
hydrothermalism and metamorphism, as a result of which only the most inmobiles
elements have been considered. The geochemical calculations, in which the major
elements take part, have been made from values recalculated to 100 in volatile free
compositions; Fe is reported as $FeO_t$.
The geochemical dataset of the Central Iberian Zone includes 152 published
geochemical data, from which 85 are plutonic and 67 volcanic and volcaniclastic rocks
from the Ollo de Sapo Formation (Galicia, Sanabria and Guadarrama areas), and the
contact between the Central Iberian and Ossa Morena Zones (Urra Formation and
Portalegre and Carrascal granites). Other data were yielded from six volcanic rocks of
the Galicia-Trás-os-Montes Zone (Saldanha area) (Fig. 1B; Repository Data).
The dataset of the eastern Pyrenees consists of 38 samples, six of which are upper
Lower Ordovician volcanic rocks, and seven upper Lower Ordovician plutonic rocks,
together with nine Upper Ordovician volcanic and 14 Upper Ordovician plutonic rocks
(Repository Data). New data reported below include two samples of subvolcanic sills
intercalated in the pre–Sardic unconformity succession (Clariana et al., 2018; Margalef,
unpubl.; Table 1).



The study samples from the Occitan Domain comprise six metavolcanites, four from
the Larroque volcanosedimentary Complex in the Albigeois and northern Montagne
Noire and two from the Mouthoumet massif (Pouclet et al., 2017) (Repository Data),
and four new samples for the Axial Zone gneisses (Table 1).
In the Sardinian dataset, 25 published analyses are selected: five correspond to the
Golfo Aranci orthogneiss (Giacomini et al., 2006), six to metavolcanites from the central
part of the island (Giacomini et al., 2006; Cruciani et al., 2013), and five to
metavolcanites and one to gneisses from the Bithia unit (Cruciani et al., 2018)
(Repository Data). Ten new analyses are added from the Monte Filau and Capo
Spartivento gneisses of the Bithia unit, and from the Punta Bianca gneisses embedded
within the migmatites of the High-grade Metamorphic complex of the Inner Zone (Table

1).

A general classification of these samples, following Winchester and Floyd (1977),
can be seen in Figure 3A–B, and the geographical coordinates of the new samples in
Table 1. For geochemical comparison (Table 2), two large groups or suites are
differentiated in order to check the similarities and differences between the magmatic
rocks, and to infer a possible geochemical trend following a palaeogeographic SW-to-
NE transect. The description reported below follows the same palaeogeographic and
chronological order.

**3.1. Furongian–to–Mid Ordovician Suite**

In the Central Iberian and Galicia-Trás-os-Montes Zones, the Furongian–to–Mid
Ordovician magmatic activity is pervasive. Their main representative is the Ollo de
Sapo Formation, which includes volcanic and subvolcanic rocks (67 samples) as well
as plutonic rocks (85 samples) (data from Murphy et al., 2006; Díez-Montes, 2007;
Montero et al., 2007, 2009; Solá, 2007; Solá et al., 2008; Talavera, 2009; Villaseca et
al., 2016). From the Parautochthon Schistose Domain of the Galicia-Trás-os Montes





Zone, six samples of rhyolite tuffs of the Saldanha Formation (Dias da Silva et al.,
2014) are selected, which share geochemical features with the Ollo de Sapo
Formation.
(i) The composition of the Ollo de Sapo-facies orthogneisses (*OG* in the figures)
ranges from potassium-rich dacite to rhyolite (60.3 < $SiO_2$ < 75 wt. %; 0.1 < $Na_2O$ < 3.9
wt. %; 3.4 < $K_2O$ < 5.9 wt. %; Figs. 3–4). This subgroup, with peraluminous A/CNK ratio
(3.1–1.0), includes samples of the Ollo de Sapo Formation from the Sanabria and
Guadarrama areas, the former dated at 472 ± 1 Ma (Díez-Montes, 2007) and the latter
between 488 ± 3 and 473 ± 8 Ma (Valverde-Vaquero and Dunning, 2000; Navidad and
Castiñeiras, 2011; Talavera et al., 2013; Villaseca et al., 2016). $\varepsilon$Nd values range from
−1.8 to −5.1, and $T_{DM}$ from 1.8 to 1.1 Ga (Montero et al., 2007, 2009; Villaseca et al.,

2016).

(ii) The composition of the leucogneisses (*LG*) ranges from potassium-rich dacite to
rhyolite (73.6 < $SiO_2$ < 75.9 wt. %; 2.7 < $Na_2O$ < 3.1 wt. %; 4.2 < $K_2O$ < 5.3 wt. %; Figs.
3–4). The A/CNK ratio is peraluminous (1.1−1.3). This subgroup includes samples from
the Guadarrama region. $\varepsilon$Nd values range from −4.9 to −5.1, and $T_{DM}$ is 4.1 Ga
(Villaseca et al., 2016). These samples display erroneous $T_{DM}$ values in two of the three
considered samples, with high $^{147}Sm/^{144}Nd$ ratios (> 0.13), a character relatively
common in felsic rocks (DePaolo, 1988; Martínez et al., 2011).
(iii) The composition of the granites (*GRA*) ranges from potassium-rich dacite to
rhyolite (64.6 < $SiO_2$ < 77 wt. %; 0.5 < $Na_2O$ < 4.8 wt. %; 2.5 < $K_2O$ < 6.3 wt. %; Figs.
3−4). The A/CNK ratio is peraluminous (1.8−1.0).This subgroup includes samples from
the northeastern Central System, Sanabria, Miranda do Douro and the western Central
Iberian Zone. The age of the involved metagranites is 487 ± 4 Ma (Montero et al.,
2009) and 488 ± 6 Ma (Díez Montes, 2007); 473 ± 3 Ma (Talavera, 2009) and 496 ± 2
Ma (Zeck et al., 2007) for the Miranda do Douro metagranites; 489 ± 5 Ma for the
Vitigudino metagranites; 486 ± 6 for the Fermoselle metagranites; and 471 ± 7 Ma for
the Ledesma metagranite (Talavera, 2009). In the southern Central Iberian Zone, the





Carrascal metagranite has been dated between 479 to 486 Ma (Solá, 2007) and the
Portalegre metagranite between 482 ± 4 and 492 ± 3 Ma (Solá, 2007). $\varepsilon$Nd values
range from +2.6 to −5.2, and $T_{DM}$ from 0.90 to 3.6 Ga (Montero et al., 2007; Solá, 2007;
Talavera, 2009).
(iv) The composition of the volcanic rocks (*VOL*) ranges from andesite to rhyolite
(64.6 < $SiO_2$ < 79.3 wt. %; 0.1 < $Na_2O$ < 3.2 wt. %; 2.2 < $K_2O$ < 6.3 wt. %; Figs. 3–4).
The A/CNK ratio is peraluminous (2.7−1.1). This subgroup includes samples from the
Saldanha Formation in the Galicia-Trás-os-Montes Zone, the metavolcanic rocks of the
Ollo de Sapo Formation in the Sanabria region and the Urra Formation. $\varepsilon$Nd values
range from −1.6 to −5.5, and $T_{DM}$ from 1.7 to 1.3 Ga (Montero et al., 2007; Solá, 2007).
(v) The composition of the San Sebastián orthogneisses (*OSS*) is rhyolitic (73.8 <
$SiO_2$ < 75.4 wt. %; 2.5 < $Na_2O$ < 3.1 wt. %; 4.9 < $K_2O$ < 5.4 wt. %; Figs. 3–4). The
A/CNK ratio is peraluminous (1.2−1.1). The San Sebastián orthogneisses are located
in the Sanabria region, on the northern Central Iberian Zone, and are dated at 465 ± 10
Ma (Lancelot et al., 1985) and 470 Ma (Talavera, 2009). They display weakly positive
$\varepsilon$Nd values (−0.0 to −4.0), and $T_{DM}$ from 1.6 to 1.2 Ga (Talavera, 2009). This subgroup
is mainly characterized by its alkaline character.
In the eastern Pyrenees, an Early−Mid Ordovician magmatic activity gave rise to the
intrusion of voluminous (about 500−3000 m in size) aluminous granitic bodies, encased
into the Canaveilles and Jujols beds (Álvaro et al., 2018; Casas et al., 2019). They
constitute the protoliths of the large orthogneissic laccoliths that form the core of the
domal massifs scattered throughout the backbone of the Pyrenees. Rocks of the
Canigó, Roc de Frausa and Albera massifs have been taken into account in this work,
in which volcanic rocks of the Pierrefite and Albera massifs, and the so-called *G3*
orthogneisses by Guitard (1970) are also included. All subgroups vary compositionally
from subalkaline andesite to rhyolite, as illustrated in the Pearce's (1996) diagram of
Figure 4 (data compiled from Vilà et al., 2005; Castiñeiras et al., 2008b; Liesa et al.,
2011; Navidad et al., 2018).





Although most rocks in this area are acidic, it is remarkable the presence of minor
mafic bodies (Cortalet and Marialles metabasites, not studied in this work), which could
indicate a mantellic connection with parental magmas during the Mid and Late
Ordovician. As well, it should be noted that there are no andesitic rocks in the area.
(vi) The composition of the ocelar orthogneisses (*G2 sensu* Guitard, 1970) ranges
from dacite to rhyolite ($68.3 < SiO_2 < 73.6$ wt. %; $3.2 < Na_2O < 3.9$ wt. %; $2.5 < K_2O < 4.4$
wt. %; Fig. 4). The age of this subgroup, with a peraluminous A/CNK ratio (1.2−1.1),
ranges from 476 to 462 Ma, $\varepsilon$Nd values from −4.4 to −3.0, and $T_{DM}$ from 1.20 to 1.44
Ga (Vilà et al., 2005; Castiñeiras et al., 2008b; Liesa et al., 2011; Navidad et al., 2018).
(vii) The composition of the *G3* orthogneisses correspond to a potassium-rich dacite
($68.4 < SiO_2 < 73.5$ wt. %; $2.4 < Na_2O = 2.9$ wt. %; $K_2O = 4.4$ wt. %; Fig. 4). The A/CNK
ratio is peraluminous (1.2). These rocks are dated at 463 ± 1 Ma (Navidad et al., 2018).
$\varepsilon$Nd value is −4.2 and $T_{DM}$ is 1.33 Ga (Navidad et al., 2018).
(viii) The composition of the volcanic rocks (*V1*) is of a sodium-rich rhyolite ($68.4 <$
$SiO_2 < 73.5$ wt. %; $2.4 < Na_2O < 7.88$ wt. %; $1.27 < K_2O < 3.2$ wt. %; Fig. 4). The
A/CNK ratio is peraluminous (2.0−1.1) (Calvet et al., 1988; Liesa et al., 2011). This
subgroup includes samples from the Pierrefite Formation and the Albera massif. The
latter has been dated from 465 ± 4 to 472 ± 3 Ma (Liesa et al., 2011). $\varepsilon$Nd value ranges
between −5.1 and −2.6, and TDM between 1.6 and 1.7 Ga (Liesa et. al., 2011;
unnpublised data).
In the Occitan Domain, six samples of the Larroque volcanosedimentary Complex
(Early Tremadocian in age) consist of basin floors and subaerial explosive and effusive
rhyolites (Pouclet et al., 2017). The porphyroclastic rocks of the Larroque metarhyolites
were sampled in the Saint-Géraud and Larroque areas from the Saint-Sernin-sur-
Rance nappe and the Saint-André klippe above the Saint-Salvi-de-Carcavès nappe
(Pouclet et al., 2017).





(ix) The composition of the Occitan volcanic rocks (*VOL-OD*) ranges from
potassium-rich dacite to rhyolite (66.7 < $SiO_2$ < 75.6 wt. %; 0.6< $Na_2O$ < 3.7 wt. %; 2.3
< $K_2O$ < 9.3 wt. %; Fig. 4). The A/CNK ratio is peraluminous (2.4−1.3).
In the Middle Ordovician rocks of Sardinia, 11 samples are selected, five of which
correspond to orthogneisses of the Aranci Gulf, in the Inner Zone of the NE island
(Giacomini et al., 2006), completed with six volcanic rocks of the External Zone
(Giacomini et al., 2006; Cruciani et al., 2018).
(x) The composition of the Sardinian orthogneisses (*OG-SMO*) corresponds to a
potassium-rich rhyolite (74 < $SiO_2$ < 67.2 wt. %; 2.6 < $Na_2O$ <3.8 wt. %; 2.3 < $K_2O$ < 5.8
wt. %; Fig. 4). These rocks, with a peraluminous A/CNK ratio (1.26–1.11), have been
dated at 469 ± 1 Ma (Giacomini et al., 2006).
(xi) Finally, the composition of the Sardinian volcanic rocks (*VOL-SMO*) ranges
from potassium-rich dacite to rhyolite (67.6 < $SiO_2$ < 76.7 wt. %; 1.9 < $Na_2O$ < 4.7 wt.
%; 2.9 < $K_2O$ < 5.4 wt. %; Fig. 4). The age of these rocks vary between 464 ± 1 Ma
(Giacomini et al., 2006) and 462 ± 4.3 Ma (Cruciani et al., 2018). The A/CNK ratio is
peraluminous (2.02−1.22).

**3.2 Upper Ordovician Suite**

In the eastern Pyrenees, four Upper Ordovician subgroups are distinguished based on
their field occurrence and geochemical and geochronological features: the *G1*-type
orthogneisses *sensu* Guitard (1970); the Cadí and Casemí orthogneisses and the
metavolcanic rocks that include the Ribes de Freser rhyolites; the Els Metges volcanic
tuffs; and the rhyolites from Andorra and Pallaresa areas. Clariana et al. (2018) have
dated the latter rhyolites at 453.6 ± 1.5 Ma.
(i) The composition of the *G1*-type orthogneisses ranges from potassium-rich dacite
to rhyodacite (73.45 < $SiO_2$ < 76.42 wt. %; 2.64 < $Na_2O$ < 3.13 wt. %; 4.73 < $K_2O$ < 5.27
wt. %; Fig. 4). The A/CNK ratio is peraluminous (1.24–1.16). These rocks have been



dated at 457 ± 1 Ma (Navidad et al., 2018). $_\varepsilon$Nd value ranges between –5.3 and –3.1,
and $T_{DM}$ between 1.47 and 2.72 Ga (Martínez et al., 2011; Navidad et al., 2018).
(ii) The *CADÍ* orthogneisses show a potassium-rich dacite to rhyodacite
composition ($SiO_2$ = 69.38 wt. %; $Na_2O$ = 3.03 wt. %; $K_2O$ = 4.05 wt. %; Fig. 4). The
A/CNK ratio is peraluminous (1.19). The age of this subgroup is 456.1 ± 4.8 Ma (Casas
et al., 2010). $Ɛ_{Nd}$ value is –4.1 and $T_{DM}$ is 1.47 Ga (Navidad et al., 2010).
(iii) The composition of the *CASEMÍ* orthogneisses ranges from potassium-rich
dacite/rhyodacite to rhyolite (71.87 < $SiO_2$ < 76.03 wt. %; 1.82 < $Na_2O$ < 4.02 wt. %;
3.24 < $K_2O$ < 6.30 wt. %) (Fig. 4). The A/CNK ratio is peraluminous (1.24–0.94). The
age ranges between 451.6 ± 4.8 Ma and 445.9 ± 4.8 Ma (Casas et al., 2010). $Ɛ_{Nd}$ value
ranges between –3.6 and –1.3, and $T_{DM}$ between 1.27 and 2.63 Ga (Navidad et al.,

534   2010).

(iv) The composition of the Pyrenean volcanic rocks (*V2*) is the most variable,
ranging from andesite to dacite/rhyodacite (62.98 < $SiO_2$ < 86.06 wt. %; 0.05 < $Na_2O$ <
5.98 wt. %; 0.63 < $K_2O$ < 4.33 wt. %; Fig. 4). The A/CNK ratio is peraluminous (3.63–
1.04). This subgroup includes the metarhyolites of Ribes de Freser, Andorra (dated at
457 ± 1.5 Ma), Pallaresa (453.6 ± 1.5 Ma) and Els Metges (455.2 ± 1.8 Ma, Navidad et
al., 2010). $_\varepsilon$Nd ranges between –5.1 and –2.6, and $T_{DM}$ between 1.62 and 1.71 Ga
(Navidad et al., 2010; Martínez et al., 2011).
(v) In the Occitan Domain, four new samples (*OG-OD*) of orthogneisses from
Gorges d' Heric (Caroux massif), S of Mazamet (Nore massif), S of Rouairoux (Agout
massif) and Le Vintrou are analyzed. The composition of the orthogneisses (*OG-OD*)
ranges from potassium-rich dacite to rhyolite (67.4 < $SiO_2$ < 73.9 wt. %; 2.8 < $Na_2O$ <
3.3 wt. %; 4.0 < $K_2O$ < 4.7 wt. %; Fig. 4). The A/CNK ratio is peraluminous (1.29–1.20).
The orthogneisses of Gorges d' Heric have been dated at 450 ± 3 Ma (Roger et al.,
2004). $_\varepsilon$Nd ranges between –3.5 and –4.0, and TDM between 1.8 and 1.4 Ga.
(vi) Fourteen samples are selected from the Upper Ordovician of Sardinia. Nine of
them correspond to orthogneisses of the External Zone (*OG-SUD*, eight samples are





new data and one taken from Cruciani et al., 2018), and five samples to volcanic rocks
from the Nappe Zone (*VOL-SUD*) (Cruciani et al., 2018). The composition of the
orthogneisses ranges from potassium-rich dacite/rhyodacite to rhyolite (72.1 < $SiO_2$ <
76.6 wt. %; 1.6 < $Na_2O$ < 3.3 wt. %; 4.8 < $K_2O$ < 7.8 wt. %; Fig. 4). The A/CNK ratio is
peraluminous (1.1–1.3). This subgroup has been dated at 464 ± 1 Ma (Giacomini et al.,
2006), and includes samples from Capo Spartivento, Cuile Culurgioni, Tuerredda,
Monte Filau and Monte Settiballas. $\epsilon_{Nd}$ value ranges from –1.6 to –3.3, and $T_{DM}$ from
1.2 to 4.2 Ga. The composition of the associated volcanic rocks ranges from
potassium-rich dacite to rhyodacite (70.7 < $SiO_2$ < 76.7 wt. %; 1.6 < $Na_2O$ < 3.3 wt. %;
4.8 < $K_2O$ < 7.8 wt. %; Fig. 4). The A/CNK ratio is peraluminous (1.1–1.3). This
subgroup includes samples of the Truzzulla Formation at Monte Grighini.

**4. Geochemical framework**

A geochemical comparison between the Furongian–Ordovician felsic rocks of all the
above-reported groups offers the opportunity to characterize the successive sources of
crustal-derived melts along the south-western European margin of Gondwana.

The geochemical features point to a predominance of materials derived from the

melting of metasedimentary rocks, rich in $SiO_2$ and $K_2O$ (average $K_2O/Na_2O$ = 2.25)
and peraluminous (0.4 < $C_{norm}$ < 4.5 and 0.94 < A/CNK > 3.12), with only three samples
with A/CNK <1 (samples 100786 of the Casemí subgroup, and T26 and T27 of the San
Sebastián subgroup).

The result of plotting the REE content vs. average values of continental crust

(Rudnick and Gao, 2004; Fig. 5) yields a flat spectra and a base level shared by most
of the considered groups. The total content in REE is moderate to high (average REE =
176 ppm, ranging between 482.2 and 26.0 ppm; Fig. 6), with a maximum in the
subgroup of the Middle Ordovician volcanic rocks from Sardinia (average REE = 335
ppm, *VOL-SMO*), and with LREE values more fractionated than HREE ones, and



negative anomalies of Eu, which would indicate a characteristic process of magmatic
evolution with plagioclase fractionation. These features are common in peraluminous
granitoids.

All subgroups display similar chondritic normalized REE patterns (Fig. 6), with an

enrichment in LREE relative to HREE, which should indicate the involvement of crustal
materials in their parental magmas. Nevertheless, some variations can be highlighted,
such as the lesser fractionation in REE content of some subgroups. These are the
leucogneisses from the Iberian massif (*LG*, $La/Yb_n$ = 2.01), the Upper Ordovician
orthogneisses from Sardinia (*OG-SUO*, $La/Yb_n$ = 2.94), the Casemí orthogneisses
($La/Yb_n$ = 4.42) and the Middle Ordovician volcanic rocks from Sardinia (*OG-SUO*,
$La/Yb_n$ = 2.94). This may be interpreted as a greater degree of partial fusion in the
origin of their parental magmas (Rollinson, 1993).

There are three geochemical groups displaying $(Gd/Yb)_n$ values > 2, and $(La/Yb)_n$

values $\geq$ 9. These groups are *OSS* (Central Iberian Zone), *VOL-OD* (Occitan Domain)
and *G1* (Pyrenees), and share higher alkalinity features.

Some *V1* rocks from the Pyrenees (Pierrefite Formation) show no negative

anomalies in Eu. Their parental magmas could have been derived from deeper origins
and related to residual materials of the lower continental crust, in areas of production of
K-rich granites (Taylor and McLenan, 1989).

The spider diagrams (Fig. 7), however, exhibit strong negative anomalies in Nb, Sr

and Ti, which indicate a distinct crustal affiliation (Díez-Montes, 2007). Only the San
Sebastián orthogneisses (*OSS*) show distinct discrepancies in respect of the remaining
samples from the Ollo de Sapo Formation. They display lower negative anomalies in
Nb and a more alkaline character by comparison with the rest of the Ollo de Sapo
rocks, which point to alkaline affinities and greater negative anomalies in Nb.

Despite some small differences in the chemical ranges of some major elements,

most felsic Ordovician rocks from the Iberian massif (Central Iberian and Galicia-Trás-
os Montes Zones), eastern Pyrenees, Occitan Domain and Sardinia share a common





chemical pattern. The Lower–Middle Ordovician rocks of the eastern Pyrenees show
less variation in the content of Zr and Nb (Fig. 7B). The volcanic rocks of these groups
show a different REE behaviour, which would indicate different sources. Two groups
are distinguished in Figure 6, one with greater enrichment in REE and negative
anomaly of Eu, and another with lesser content of HREE and without Eu negative
anomalies.

Figure 8 illustrates how the average of all the considered groups approximates the

mean values of the Rudnick and Gao's (2003) Upper Continental Crust. In this figure,
small deviations can be observed, some of them toward LCC values and others toward
BCC, indicating variations in their parental magmas but with quite similar spectra.
Overall chondrite-normalized patterns are close to the values that represent the upper
continental crust, with slight enrichments in the Th/Nb, Th/La and Th/Yb ratios.

Finally, in the Occitan volcanic rocks (*VOL-OD*) the rare earth elements are

enriched and fractionated (33.2 ppm < La < 45.6 ppm; 11.2 < La/Yb < 14.5). The upper
continental crust normalized diagram exhibits negative anomalies of Ti, V, Cr, Mn and
Fe associated with oxide fractionation, of Zr and Hf linked to zircon fractionation, and of
Eu related to plagioclase fractionation. The profiles are comparable to the Vendean
Saint-Gilles rhyolitic ones. The Th vs. Rb/Ba features are also similar to those of the
Saint-Gilles rhyolites, and the Iberian Ollo de Sapo and Urra rhyolites (Solá et al.,
2008; Díez Montes et al., 2010).

**4.1 Inferred tectonic settings**

In order to clarify the evolution of geotectonic environments, the data have been
represented in different geotectonic diagrams. The Zr/$TiO_2$ ratio (Lentz, 1996; Syme,
1998) is a key index of compositional evolution for intermediate and felsic rocks. In the
Syme diagram (Fig. 9), most rocks from the Central Iberian Zone represent a
characteristic arc association, although there are some contemporaneous samples





characterized by extensional-related values (Zr/Ti = 0.10, *LG*). The rocks of the
Middle–Ordovician San Sebastián orthogneisses (*OSS*) show values of Zr/Ti = 0.08,
intermediate between extensional and arc conditions. This could be interpreted as a
sharp change in geotectonic conditions toward the Mid Ordovician (Fig. 9A). For a
better comparison, the samples of the San Sebastián orthogneisses (*OSS*) and the
granites (*GRA*) have been distinguished with a shaded area in all the diagrams, since
they have slightly different characteristics to the rest of the samples from the Ollo de
Sapo group. The samples *G1* (Pyrenees) and *VOL* (Central Iberian Zone) broadly
share similar values, as a result of which, the three latter groups (*OSS*, *G1* and *VOL*)
arrange following a good correlation line. The same trend seems to be inferred in the
eastern Pyrenees (Fig. 9B), where the Middle Ordovician subgroups display arc
features, but half of the Upper Ordovician subgroups show extensional affinities (*G1*
and Casemí orthogneisses). In the case of the Occitan orthogneisses (Fig. 9C), they
show arc characters, which contrast with the contemporaneous volcanic rocks
displaying extensional values with Zr/Ti = 0.10. This disparity between plutonic and
volcanic rocks could be interpreted as different conditions for the origin of these
magmas. In Sardinia (Fig. 9D), the same evolution from arc to extensional conditions is
highlighted for the Upper Ordovician samples, although some Middle Ordovician
volcanic rocks already shared extensional patterns (Zr/Ti = 0.09). In summary, there
seems to be a geochemical evolution in the Ordovician magmas grading from arc to
extensional environments.
In the Nb–Y tectonic discriminating diagram of Pearce et al. (1984) (Fig. 10), most
samples plot in the volcanic arc-type, though some subgroups project in the whitin-
plate and anomalous ORG. The majority of samples display very similar Zr/Nb and
Nb/Y ratios, typical of island arc or active continental margin rhyolites (Díez-Montes et
al., 2010). Only some samples plot separately: *OSS* samples with highest Nb contents
(>20 ppm), and some volcanic rocks of the Occitan Domain (average Nb =16.87 ppm).
In the eastern Pyrenees, the Middle Ordovician rocks plot in the volcanic arc field,





whereas the Upper Ordovician ones point in the ORG type, except the Casemí
samples. This progress of magmatic sources agrees with the evolution seen in Figure
9. In the Ocitan Domain, *VOL-OD* samples share values with those of the San
Sebastián orthogneiss, while *OG-OD* shares values with those of *OG* from the Central
Iberian Zone.
The Zr vs. Nb diagram (Leat et al., 1986; modified by Piercey, 2011) (Fig. 11)
illustrates how magmas evolved toward richer values in Zr and Nb, which is consistent
with what it is observed in the Syme diagram (Fig. 9). Figure 11A documents how most
samples show a general positive trend where two groups are distinguished. These
different groups correspond to the *OSS* and Portalegre granites, highlighted in the
figure. The two groups indicate a tendency toward alkaline magmas. In the rest of the
diagrams, the groups from the Central Iberian Zone are projected in blue. Some
samples, such as the Pyrenean *G1*, some Occitan *VOL-OD* samples and some
Sardinian *OG-UOS* samples share the same affinity, clearly distinguished from the
general geochemical trend exhibited by the Central Iberian Zone.
After plotting the data in a Zr vs. Ga/Al diagram (Whalen et al., 1987) (Fig. 12), the
samples depict an intermediate character between alkaline and I&S. In the Central
Iberian Zone, samples from the San Sebastián orthogneisses and Portalegre granites
show characters of A-type granites, while the remaining samples display affinities of
I&S-type granites. For the Central Iberian Zone, a clear magmatic shift toward more
extensional geotectonic environments is characterized. For the eastern Pyrenees, we
find the same situation than for the Central Iberian Zone, with a magmatic evolution
toward A-granite type characteristics, indicating more extensional geotectonic
environments. In the Occitan Domain, the samples show a clear I&S character. In the
Sardinian case, the same seems to happen as in the Central Iberian Zone: the Upper
Ordovician orthogneisses suggest a more extensional character.
In summary, all the reported diagrams point to a magmatic evolution through time,
grading from arc to extensional geotectonic environments (with increased Zr/Ti ratios)



and to granite type-A characters. This geotectonic framework is consistent with that
illustrated in Figure 9. The geochemical characters of these rocks show a rhyodacite to
dacite composition, peraluminous and calc-alkaline K-rich character, and an arc-
volcanic affinity for most of samples, but without intermediate rocks associated with
andesitic types. Hence a change in time is documented toward more alkaline magmas.

**4.2 Interpretation of $_\varepsilon$Nd values**

$_\varepsilon$Nd values are useful to interpret the nature of magmatic sources. Most samples of the
above-reported groups show no meaningful differences in isotopic $_\varepsilon$Nd values, and
$Nd_{CHUR}$ model ages (Fig. 13). Some exceptions are related to granites from the
southern Central Iberian Zone, which display positive values (from +2.6 to −2.4) and
$T_{DM}$ values from 0.90 to 3.46 Ga. This feature could be interpreted as a more primitive
nature of their parental magmas, even though the samples with highest $T_{DM}$ values are
those that have higher $^{147}Sm/^{144}Nd$ ratios (> 0.16; Table 1). On the other hand, very
high values of the $^{147}Sm/^{144}Nd$ ratio (> 0.13) could indicate post-magmatic hydrothermal
alteration of the orthogneissic protoliths, as pointed out by Martínez et al. (2011).
These are the case for samples from the Central Iberian Zone, VI-3 sample
(Leucogneisses subgroup) and PORT2 and PORT15 of the Granites subgroup; as well
as in the eastern Pyrenees, 99338 sample (G1 subgroup) and 100786 sample (Casemí
subgroup). In Sardinia, CS5, CS8 and CC5 samples of the Upper Ordovician
Orthogneisses subgroup show the highest values in $T_{DM}$ (Table 2; Fig. 13).
The volcanic rocks of the Central Iberian Zone display some differences following a
N-S transect, being $\varepsilon_{Nd}$ values more negative in the north ($_\varepsilon$Nd: −4.0 to −5.0) than in
the south ($_\varepsilon$Nd: −1.6 to −5.5). The isotopic signature of the Urra volcaniclastic rocks is
compatible with magmas derived from young crustal rocks, with intermediate to felsic
igneous compositions (Solá et al., 2008). The volcanic rocks of the northern Central
Iberian Zone could be derived from old crustal rocks (Montero et al., 2007). The





isotopic composition of the granitoids from the southern Central Iberian Zone has more
primitive characters than those of the northern Central Iberian Zone, suggesting
different sources for both sides (Talavera et al., 2013). *OSS* shows lower inheritance
patterns, more primitive Sr–Nd isotopic composition than other rocks of the Ollo de
Sapo suite, and an age some 15 m.y. younger than most meta-igneous rocks of the
Sanabria region (Montero et al., 2009), likely reflecting a greater mantle involvement in
its genesis (Díez-Montes et al., 2008).
According to Talavera et al. (2013), the Cambro–Ordovician rocks of the Galicia-
Trás-os-Montes Zone schistose area and the magmatic rocks of the northern Central
Iberian Zone are contemporary. Both metavolcanic and metagranitic rocks almost
share the same isotopic compositions.
The Upper Ordovician orthogneisses from the Occitan Domain show very little
variation in $\varepsilon$Nd values (–3.5 to –4.0), typical of magmas derived from young crustal
rocks. The variation in TDM values is also small (1.4 to 1.8 Ga) indicating short crustal
residence times.
In Sardinia, $\varepsilon$Nd values present a greater variation (–1.6 to –3.3), but they are also
included in the typical continental crustal range. As noted above, anormal TDM values
(between 1.2 to 4.5 Ga) may be due to post-magmatic hydrothermal alteration
processes.

**5. Geodynamic scenario**

In the Iberian Massif, the Ediacaran–Cambrian transition was marked by
paraconformities and angular discordances indicating the passage from Cadomian
volcanic arc to rifting conditions. The axis of the so-called Ossa-Morena Rift lies along
the homonymous Zone (Quesada, 1991; Sánchez-García et al., 2003, 2008, 2010)
close to the remains of the Cadomian suture (Murphy et al., 2006). Rifting conditions
were accompanied by a voluminous magmatism that changed from peraluminous acid





to bimodal (Sánchez-García et al., 2003, 2008, 2016, 2019). Some authors (Álvaro et
al., 2014; Sánchez-García et al., 2019) propose that this rift resulted from a SW-to-NE
inward migration, toward innermost parts of Gondwana, of rifting axes from the Anti-
Atlas in Morocco to the Ossa-Morena Zone in the Iberian Massif. According to this
proposal the rifting developed later (in Cambro–Ordovician times) in the Iberian,
Armorican and Bohemian massifs.
The Furongian–Ordovician transition to drifting conditions is associated, in the
Iberian Massif, Occitan Domain, Pyrenees and Sardinia, with a stepwise magmatic
activity contemporaneous with the record of the Toledanian and Sardic unconformities.
These, related to neither metamorphism nor penetrative deformations, are linked to
uplift, erosion and irregularly distributed mesoscale deformation that gave rise to
angular unconformities up to 90º. The time span involved in these gaps is similar (22
m.y. in the Iberian Massif, 16–23 m.y. in the Pyrenees and 18 m.y. in Sardinia). This
contrasts with the greater time span displayed by the magmatic activity (30–45 m.y.),
which started before the unconformity formation (early Furongian in the Central Iberian
Zone vs. Floian in the Pyrenees, Occitan Domain and Sardinia), pursuit during the
unconformity formation (Furongian and early Tremadocian in the Central Iberian Zone
vs. Floian–Darriwilian in the Pyrenees, Occitan Domain and Sardinia), and ended
during the sealing of the uplifted and eroded palaeorelief (Tremadocian–Floian
volcaniclastic rocks at the base of the Armorican Quartzite in the Central Iberian Zone
vs. Sandbian–Katian volcanic rocks at the lowermost part of the Upper Ordovician
successions in the Pyrenees, Occitan Domain and Sardinia; Gutiérrez-Alonso et al.,
2007, 2016; Navidad et al., 2010; Martínez et al., 2011; Álvaro et al., 2016; Martí et al.,
2019). In the Pyrenees, Upper Ordovician magmatism and sedimentation coexist with
normal faults controlling marked thickness changes of the basal Upper Ordovician
succession and cutting the lower part of this succession, the Sardic unconformity and
the underlying Cambro–Ordovician sequence (Puddu et al., 2018, 2019).



*Toledanian Phase*

The Early Ordovician (Toledanian) magmatism of the Central Iberian Zone evolved to a
typical passive-margin setting, with geochemical features dominated by acidic rocks,
peraluminous and rich in K, and lacking any association with basic or intermediate
rocks. Some of the orthogneisses of the Galicia-Trás-os-Montes Zone basal and
allochthonous complex units share these same patterns. This fact has been interpreted
by some authors as a basin environment subject to important episodes of crustal
extension (Martínez-Catalán et al., 2007; Díez-Montes et al., 2010). In contrast,
Villaseca et al. (2016) interpreted this absence as evidence against rifting conditions,
though the absence of contemporary basic magmatism may be explained by the partial
fusion of a thickened crust, through recycling of Neoproterozoic crustal materials. The
thrust of a large metasedimentary sequence could generate dehydration and
metasomatism of the rocks above this sequence, triggering partial fusion at different
levels, although the increase in peraluminosity with the basicity of the ortogneisses is
against any AFC process involving mantle materials. However, this increase in
peraluminosity with the basicity has not been revealed in the samples studied above.
Following Villaseca et al.'s (2016) model, a flat subduction of the southern part of the
Central Iberian Zone would have taken place under its northern prolongation, whereas
the reflection of such a subduction is not evident in the field. The calc-alkaline signature
of this magmatism has also been taken into account as proof of its relationship with
volcanic-arc environments (Valverde-Vaquero and Dunning, 2000). However, calc-
alkaline features may be also interpreted as a result of a variable degree of continental
crustal contamination and/or previously enriched mantle source (Sánchez-García et al.,
2003, 2008, 2016, 2019; Díez-Montes et al., 2010). Finally, other granites not
considered here of Tremadocian age have been reported in the southern Central
Iberian Zone, such as the Oledo massif and the Beira Baixa-Central Extremadura,
which display a I-type affinity (Antunes et al., 2009; Rubio Ordóñez et al., 2012). These



granites could represent different sources for the Ordovician magmatism in the Central
Iberian Zone.
Sánchez-García et al. (2019) have proposed that the anomaly that produced the
large magmatism throughout the Iberian Massif could have migrated from the rifting
axis to inwards zones and the acid, peraluminous, K-rich rocks of Mid Ordovician in
age should represent the initial stages of a new rifting pulse, resembling the
peraluminous rocks of the Early Rift Event *sensu* Sánchez-García et al. (2003) from the
Cambrian Epoch 2 of the Ossa-Morena Rift.
In the parautochthon of the Galicia-Trás-os-Montes Zone, the appearance of
tholeiitic and alkaline-peralkaline magmatism in the Mid Ordovician would signal the
first steps toward extensional conditions (Díez Fernández et al., 2012; Dias da Silva et
al., 2016). In the Montagne Noire and the Mouthoumet massifs contemporaneous
tholeiitic lavas indicate a similar change in the tectonic regimen (Álvaro et al., 2016).
This gradual change in geodynamic conditions is also marked by the appearance of
rocks with extensional characteristics in some of subgroups considered here, such as
the Central Iberian Zone (San Sebastián orthogneisses), eastern Pyrenees (Casemí
orthoneisses, and G1), volcanic rocks of the Occitan Domain, and the ortogneises and
volcanic rocks from Sardinia.

*Sardic Phase*

In the eastern Pyrenees, two peaks of magmatic activity have been currently
distinguished (Casas et al., 2019). Large Lower–Middle Ordovician peraluminous
granite bodies are known representing the protoliths of numerous gneissic bodies with
laccolithic morphologies. In the Canigó massif, the Upper Ordovician granite bodies
(protholits of Cadí, Casemí, *G1*) are encased in sediments of the Canaveilles and
Jujols groups. During this time span, there was generalized uplift and erosion that
culminated with the onset of the Sardic unconformity. The Sardic Phase was





succeeded by an extensional pulsation related to the formation of normal faults
affecting the pre–unconformity strata (Puddu et al., 2018, 2019). The volcanic arc
signature can be explain by crustal recycling (Navidad et al., 2010; Casas et al., 2010;
Martínez et al., 2011), as in the case of the Toledanian Phase in the Central Iberian
Zone, although, according to Casas et al. (2019), the Pyrenees and the Catalan
Coastal Ranges were probably fringing the Gondwana margin in a different position
than that occupied by the Iberian Massif. As a whole, the Ordovician magmatism in the
eastern Pyrenees lasted about 30 m.y., from ca 477 to 446 Ma, in a time span
contemporaneous with the formation of the Sardic unconformity (Fig. 2). Recently,
Puddu et al. (2019) proposed that a thermal doming, bracketted between 475 and 450
Ma, should have stretched the Ordovician lithosphere. The emersion and denudation of
the inherited Cambrian–Ordovician palaeorelief would have given rise to the onset of
the Sardic unconformity. According to these authors, thermal doming triggered by hot
mafic magma underplating may also be responsible for the late Early–Late Ordovician
coeval magmatic activity.
In the Occitan Domain, there was a dramatic volcanic event in early Tremadocian
times, with the uprising of basin floors and the subsequent effusion of abundant
rhyolitic activities under subaerial explosive conditions (Larroque volcanosedimentary
Complex in the Montagne Noire, and Davejean acidic volcanic counterpart in the
Mouthoumet Massif). Pouclet el al., (2017) interpreted this as a delayed Ollo de Sapo-
style outpouring where a massive crustal melting required a rather significant heat
supply. Asthenospheric upwelling leading to the interplay of lithospheric doming,
continental break-up, and a decompressionally driven mantle melting can explain such
a great thermal anomaly. The magmatic products accumulated on the mantle-crust
contact would provide enough heat transfer for crustal melting (Huppert and Sparks,
1988). Subsequently, a post–Sardic reactivation of rifting conditions is documented in
the Cabrières klippes (southern Montagne Noire) and the Mouthoumet massif. There, a
Late Ordovician fault-controlled subsidence linked to the record of rift-related tholeiites





(Roque de Bandies and Villerouge formations) were contemporaneous with the record
of the Hirnantian glaciation (Álvaro et al., 2016). Re-opening of rifting branches
(Montagne Noire and Mouthoumet massifs) was geometrically recorded as onlapping
patterns and final sealing of Sardic palaeoreliefs by Silurian and Lower Devonian
strata.
Sardinia illustrates an almost complete record of the Variscan Belt (Carmignani et
al., 1994; Rossi et al., 2009). Some plutonic orthogneises of the Inner Zone belong to
this cycle, such as the orthogneises of Golfo Aranci (Giacomini et al., 2006). Gaggero
et al. (2012) described three magmatic cycles. The first cycle is well represented in the
Sarrabus unit by Furongian–Tremadocian volcanic and subvolcanic interbeds within a
terrigenous sucession (San Vito Formation) which is topped by the Sardic
uncomformity. Some plutonic orthogneises of the Inner Zone belong to this cycle, such
as the orthogneises of Golfo Aranci (Giacomini et al., 2006) and the PB orthogneiss of
Punta Bianca). The second Mid–Ordovician cycle, about 50 m.y. postdating the
previous cycle, is of an arc-volcanic type with calc-alkaline affinity and acidic-to-
intermediate composition. The acidic metavolcanites are referred in the literature as
"porphyroids", which crop out in the External Nappe Zone and some localities of the
Inner Zone. The intermediate to basic derivates are widespread in Central Sardinia
(Serra Tonnai Formation). Some plutonic rocks (Mt. Filau orthogneisses and Capo
Spartivento) of the second cycle are discussed above. The third cycle consists of
alkalic meta-epiclastites interbedded in post–Sandbian strata and metabasites marking
the Ordovician/Silurian contact and reflecting rifting conditions. In this work only the first
two cycles has been considered. Giacomini et al. (2006) cite coeval mafic rocks of
felsic magmatism of Mid Ordovician age (Cortesogno et al., 2004; Palmeri et al., 2004;
Giacomini et al., 2005), although they interpret a subduction scenario of the Hun terrain
below Corsica and Sardinia in the Mid Ordovician.

*Intracrustal siliceous melts*




In this scenario, the key to generate large volumes of acidic rocks in an intraplate
context would be the existence of a lower-middle crust, highly hydrated, in addition to a
high heat flow, possibly caused by mafic magmas (Bryan et al., 2002; Díez-Montes,
2007). This could be the scenario raised by the arrival of a thermal anomaly in a
subduction-free area (Sánchez-García et al., 2003, 2008, 2019; Álvaro et al., 2016).
The formation of large volumes of intracrustal siliceous melts could act as a viscous
barrier, preventing the rise of mafic magmas within volcanic environments, and causing
the underplating of these magmas at the contact between the lower crust and the
mantle (Huppert and Sparks, 1988; Pankhurst et al., 1998; Bindeman and Valley,
2003). The cooling of these magmas could lead to crustal thickening and in this case,
the volcanic arc signature can be explained by crustal recycling (Navidad et al., 2010;
Díez-Montes et al., 2010; Martínez et al., 2011).
Sánchez-García et al. (2019) have proposed that the anomaly that produced the
large magmatism throughout the Iberian Massif could have migrated from the rifting
axis to inwards zones and the acid, peraluminous, K-rich rocks of Mid Ordovician in
age should represent the initial stages of a new rifting pulse, resembling the
peraluminous rocks of the Early Rift Event *sensu* Sánchez-García et al. (2003) from the
Cambrian Epoch 2 of the Ossa-Morena Rift. In the parautochthon of the Galicia-Trás-
os-Montes Zone, the appearance of tholeiitic and alkaline-peralkaline magmatism in
the Mid Ordovician would signal the first steps toward extensional conditions (Díez
Fernández et al., 2012; Dias da Silva et al., 2016). In the Montagne Noire and the
Mouthoumet massifs contemporaneous tholeiitic lavas indicate a similar change in the
tectonic regimen (Álvaro et al., 2016). This change in geodynamic conditions is also
marked by the appearance of rocks with extensional characteristics in some of
subgroups considered here, such as the Central Iberian Zone (San Sebastián
orthogneisses), eastern Pyrenees (Casemí orthoneisses, and G1), volcanic rocks of
the Occitan Domain, and the ortogneises and volcanic rocks from Sardinia. In the





Pyrenees, Puddu et al. (2019) proposed that a thermal doming, between 475 and 450
Ma, should have stretched the Ordovician lithosphere leading to emersion and
denudation of a Cambrian–Ordovician palaeorelief, and giving rise to the onset of the
Sardic unconformity. According to these authors, thermal doming triggered by hot mafic
magma underplating may also be responsible for the late Early–Late Ordovician coeval
magmatic activity
A major continental break-up, leading to the so-called Tremadocian Tectonic Belt,
was suggested by Pouclet et al. (2017), which initiated by upwelling of the
asthenosphere and tectonic thinning of the lithosphere. Mantle-derived mafic magmas
were underplated at the mantle-crust transition zone and intruded the crust. These
magmas provided heat for crustal melting, which supplied the rhyolitic volcanism. After
emptying the rhyolitic crustal reservoirs, the underlying mafic magmas finally rised and
reached the surface. According to Pouclet et al. (2017), the acidic magmatic output
associated with the onset of the Larroque metarhyolites resulted in massive crustal
melting requiring a rather important heat supply. Asthenospheric upwelling leading to
lithospheric doming, continental break-up, and a decompressionally driven mantle
melting can explain such a great thermal anomaly. Magmatic products accumulated on
the mantle-crust contact providing enough heat transfer for crustal melting.

**6. Conclusions**

A geochemical comparison of 231 plutonic and volcanic samples of two major suites,
Furongian–Mid Ordovician and Late Ordovician in age, and recorded in the Central
Iberian and Galicia-Trás-os-Montes Zones of the Iberian Massif and in the eastern
Pyrenees, Occitan Domain (Albigeois, Montagne Noire and Mouthoumet massifs) and
Sardinia, is made in this work. The comparison points to a predominance of materials
derived from the melting of metasedimentary rocks, peraluminous and rich in $SiO_2$ and
$K_2O$. The total content in REE is moderate to high. Most felsic rocks display similar





chondritic normalized REE patterns, with an enrichment of LREE relative to HREE,
which should indicate the involvement of crustal materials in their parental magmas.
$Zr/TiO_2$, Zr/Nb, Nb/Y and Zr vs. Ga/Al ratios, and REE and $\varepsilon$Nd values reflect
contemporaneous arc and extensional scenarios, which progressed to distinct
extensional conditions finally associated with outpouring of mafic tholeiitic-dominant
rifting lava flows. Magmatic events are contemporaneous with the formation of the
Toledanian (Furongian–Early Ordovician) and Sardic (Early–Late Ordovician)
unconformities, related to neither metamorphism nor penetrative deformation. The
geochemical and structural framework precludes a subduction scenario reaching the
crust in a magmatic arc to back-arc setting. On the contrary, it favours partial melting of
sediments and/or granitoids in a continental lower crust triggered by the underplating of
hot mafic magmas during extensional events related to the opening of the Rheic Ocean
as a result of asthenospheric upwelling.

**7. Acknowledgements**

This paper is a contribution to projects CGL2017-87631-P and PGC2018-093903-B-
C22 from Spanish Ministry of Science and Innovation.

**Data availability** - All data included in the paper and the Repository Data.

**Author contributions** - JJA, TSG and JMC: Methodology (Lead), Supervision (Lead),
Writing – Original Draft (Lead), Writing – Review & Editing (Lead); CP, ADM, ML & GO:
Methodology (Supporting), Supervision (Supporting), Writing – Original Draft
(Supporting), Writing – Review & Editing (Supporting).

**Competing interests** - No competing interests



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


**FIGURES**

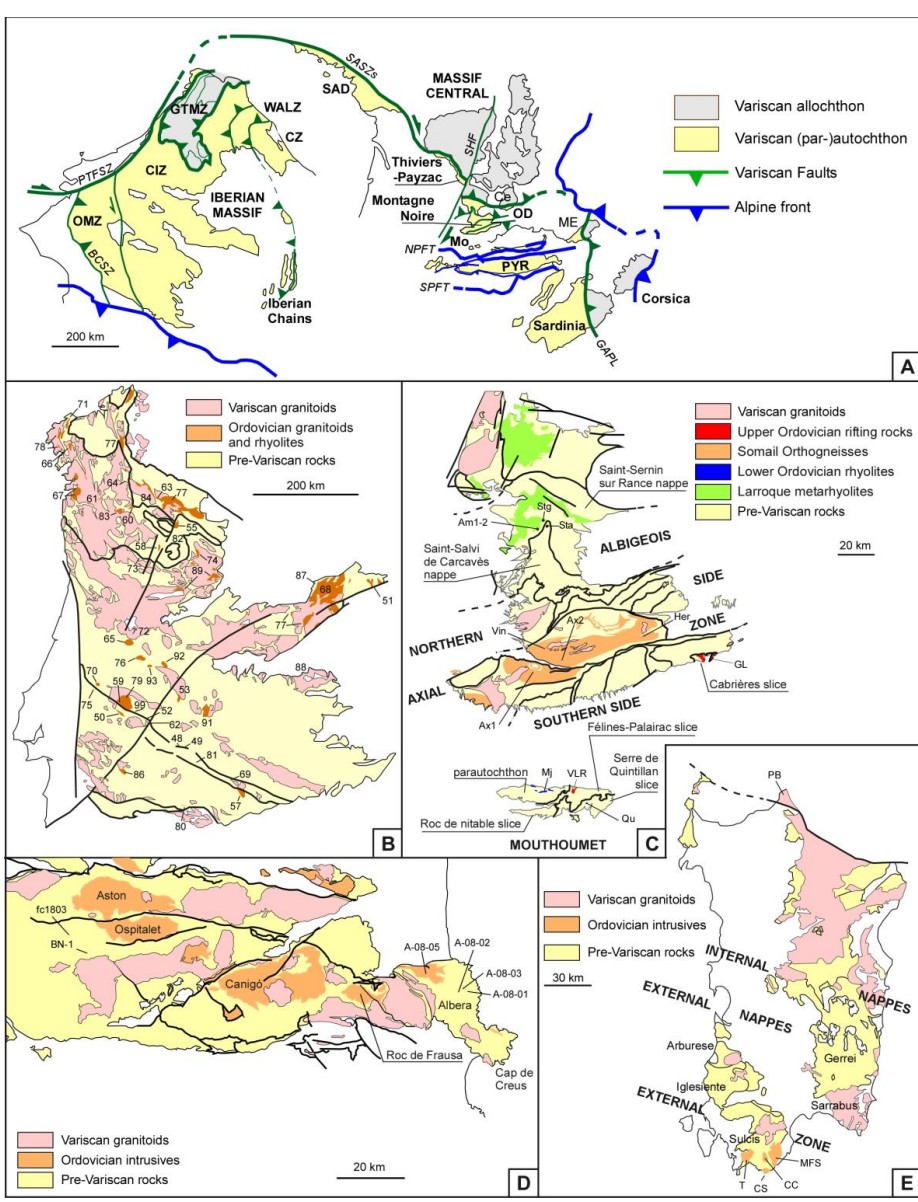


**Figure 1.** A. Reconstruction of the south-western European margin of Gondwana in
Late Carboniferous–Early Permian times; modified from Pouclet et al. (2017). B.
Setting of samples in the Central Iberian and Galicia-Trás-os-Montes zones; 48-
Aceuchal, 49- Almendralejo, 50-Alter do Chao-Alter Pedroso, 51-Antoñita, 52-





Arronches, 53- Arroyo de la Luz, 55- Bragança, 57- Cardenchosa, 58 -Carrapatas,
Facho & Valbenfeito, 59- Carrascal, 60- Carraxo, 61- Celanova-Bande, 62- Cevadais,
63- Covelo, 64- Os los Peares, 65- Fundao, 66- Galicia orthogneiss, 69- Las Minillas,
70- Maçao, 71- Malpica, 72- Manteigas, 73- Marão-Eucisia-Moncorvo, 74- Miranda do
Douro, 75- Mouriscas, 76- Oledo, 77- Ollo de Sapo, 78- Pontevedra-Sisargas, 79-
Portalegre, 80- Ribera deHuelva, 81- Rivera del Fresno, 82- Saldanha, 83- San
Mamede, 84-San Sebastián, 86- São Marcos do Campo, 87-Tenzuela, 88- Toledo
(Anatectic Dome), 89- Tormes Dome, 90- Urra, 91- Zarza de Montanchez 92- Zarza la
Mayor and 93- Zebreira; modified from Sánchez-García et al. (2019). C. Setting of
samples in the Montagne Noire and Mouthoumet massifs; Am1-2 Larroque hamlet
(Ambialet), Stg- St.Géraud  Sta- St. André, Mj- Montjoi, Qu- Quintillan, GL- Roque de
Bandies, VLR- Villerouge-Termenès, VIN- Le Vintrou, HER- Gorges d'Héric (Caroux
massif), Ax1- S Mazamet (Nore massif), Ax2 (Rou)- S Rouayroux (Agout massif);
modified from Álvaro et al. (2016). D. Setting of Pyrenean samples; modified from
Casas et al. (2019). E. Setting of Sardinian samples; CS 2,3,4,8- Spartivento Cap, T2-
Tuerreda, CC5- Cuile Culurgioni, MF1- Monte Filau, MFS1-Monte Settiballas, PB-
Punta Bianca; modified from Oggiano et al. (2010).





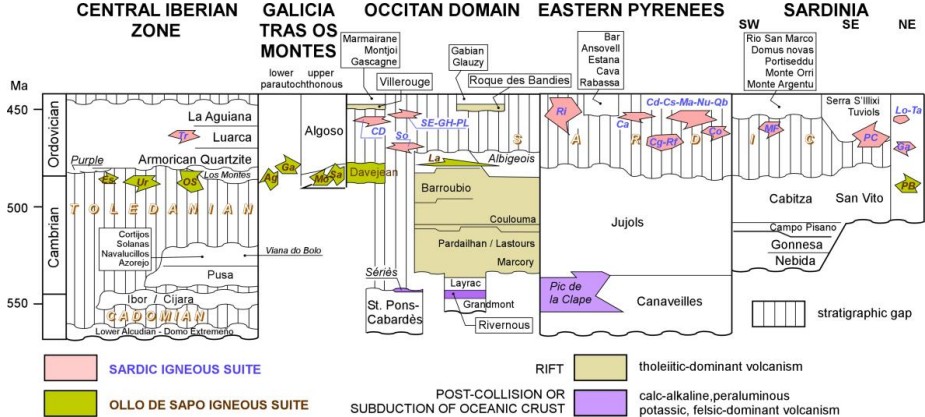


**Figure 2.** Stratigraphic comparison of the Cambro-Ordovician successions from the Central Iberian Zone, Galicia Trás-os-Montes Zone, Occitan Domain, Eastern Pyrenees and Sardinia; modified from Álvaro et al. (2014b, 2016, 2018), Pouclet et al. (2017) and Sánchez-García et al. (2019); abbreviations: *Ca* Campelles ignimbrites (ca. 455 Ma,Martí et al., 2014), *CD* Cadí gneiss (456 ± 5 Ma, Casas et al., 2010), *Cg* Canigó gneiss (472–462 Ma, Cocherie et al., 2005; Navidad et al., 2018), *Co* Cortalets metabasite (460 ± 3 Ma, Navidad et al., 2018), *Cs* Casemí gneiss (446 ± 5 and 452 ± 5 Ma, Casas et al., 2010), *Es* Estremoz rhyolites (499 Ma, Pereira et al., 2012), *Ga* Golfo Aranci orthogneiss (469 ± 3.7 Ma, Giacomini et al., 2006), *GH* Gorges d'Heric orthogneiss (450 ± 6 Ma, Roger et al., 2004), *La* Larroque Volcanic Complex, *Ma* Marialles microdiorite (453 ± 4 Ma, Casas et al., 2010), *Lo* Lodè orthogneiss (456 ± 14 Ma, Helbing and Tiepolo, 2005), *MF* Monte Filau-Capo Spartivento orthogneiss (449 ± 6 Ma, Ludwing and Turi, 1989; 457.5 ± 0,3 and 458.2 ± 0.3 Ma, Pavanetto et al., 2012), *Nu* Núria gneiss (457 ± 4 Ma, Martínez et al., 2011), *OS* Ollo de Sapo rhyolites and ash-fall tuff beds (ca. 477 Ma., Gutiérrez-Alonso et al., 2016), *PL* Pont de Larn orthogneiss (456 ± 3 Ma, Roger et al., 2004), *Qb* Queralbs gneiss (457 ± 5 Ma, Martínez et al., 2011), *PB* Punta Bianca orthogneiss (broadly Furongian–Tremadocian in age), *PC* Porto Corallo dacites (465.4 ± 1.9 and 464 ± 1 Ma, Giacomini et al., 2006; Oggiano et al., 2010), *Ri* Ribes granophyre (458 ± 3 Ma, Martínez et al., 2011), *Rf* Roc



1614 de Frausa gneiss (477 ± 4, 476 ± 5 Ma, Cocherie et al., 2005; Castiñeiras et al., 2008),

1615 *So* Somail orthogneiss (471 ± 4 Ma, Cocherie et al. 2005), *SE* Saint Eutrope gneiss

1616 (455 ± 2 Ma, Pitra et al., 2012), *Ta* Tanaunella orthogneiss 458 ± 7 Ma (Helbing and

1617 Tiepolo, 2005), *Tr* Turchas and *Ur* Urra rhyolites.






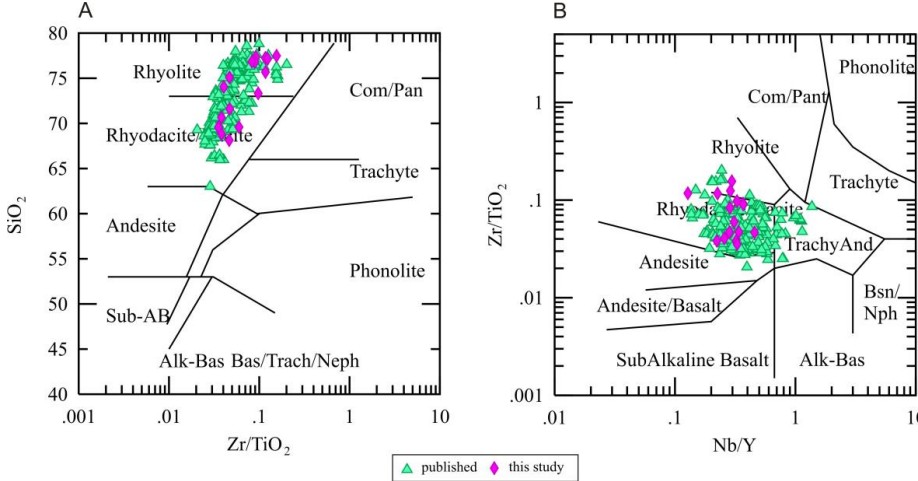



**Figure 3. SiO$_2$ vs. Zr/TiO$_2$ and Zr/TiO$_2$ vs. Nb/Y plots (Winchester and Floyd, 1977)**

**showing the composition of new samples (purple diamonds) and those taken**

**from the literature (green triangles).**






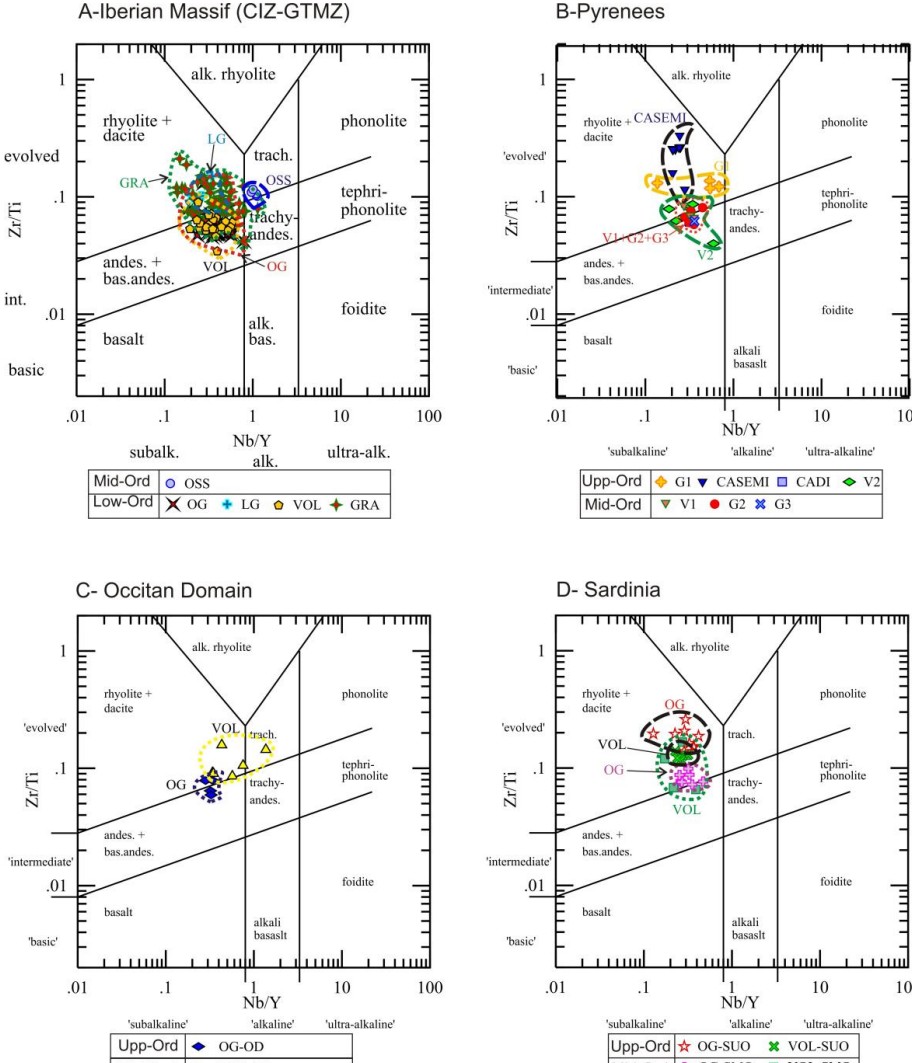

**Figure 4. Zr/Ti vs. Nb/Y discrimination diagram (after Winchester and Floyd, 1977; Pearce, 1996). A. Lower–Middle Ordovician rocks of Iberian Massif (Central Iberian and Galicia-Trás-os-Montes zones). B. Middle–Upper Ordovician rocks of the eastern Pyrenees. C) Middle Ordovician rocks of the Occitan Domain. C–D. Middle–Upper Ordovician rocks of Sardinia.**





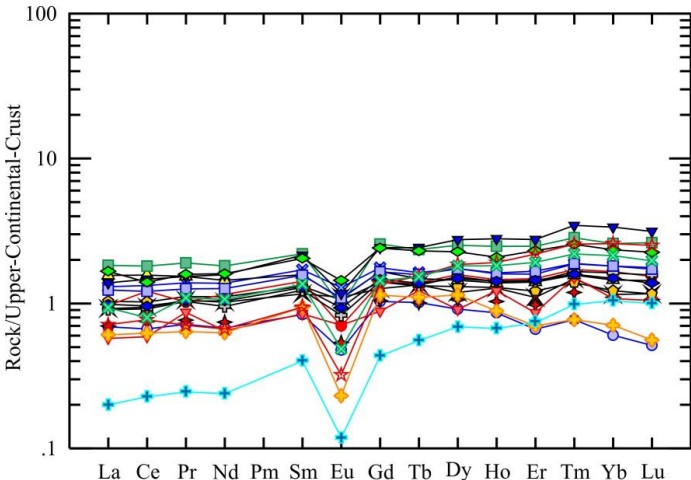


**Figure 5. Upper Crustal-normalized REE patterns (Rudnick and Gao, 2003) with**

**average values for all distinguished groups; symbols as in Figure 4.**








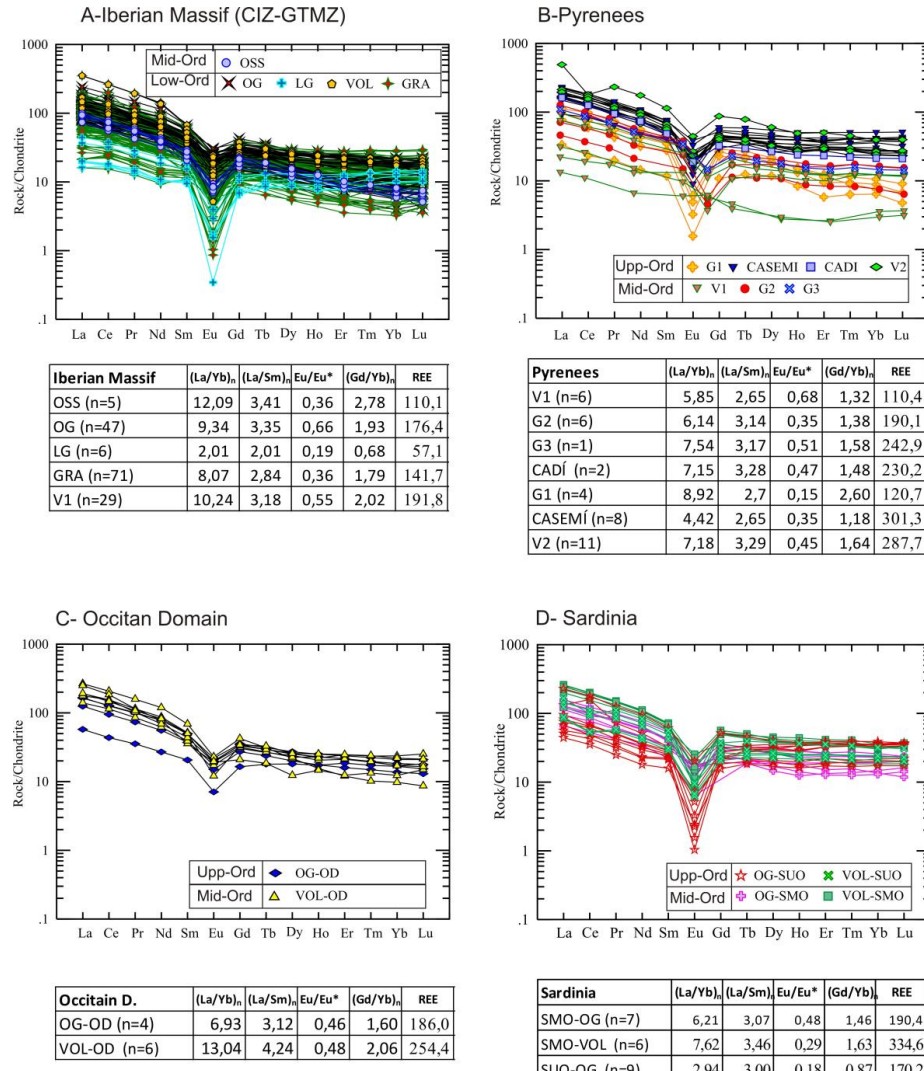


**Figure 6. Chondrite-normalized REE patterns (Sun and McDonough, 1989) for all**

**study samples.**






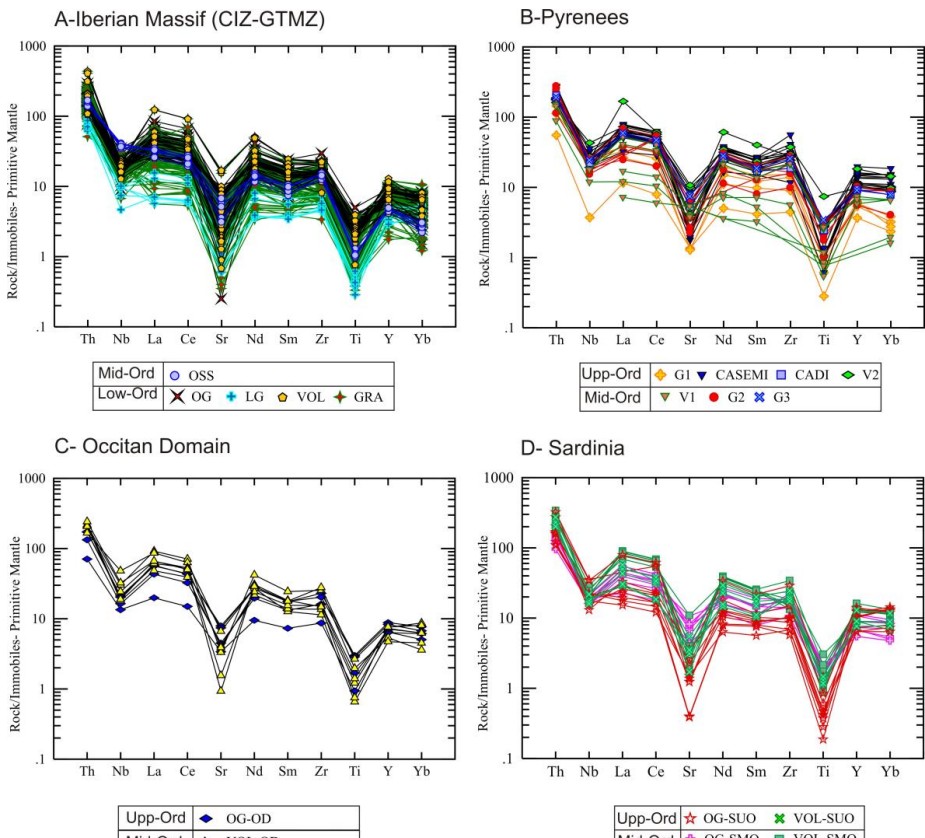

**Figure 7. Multi-element diagram normalised to Primitive Mantle of Palme and O'Neill (2004) for all study samples.**



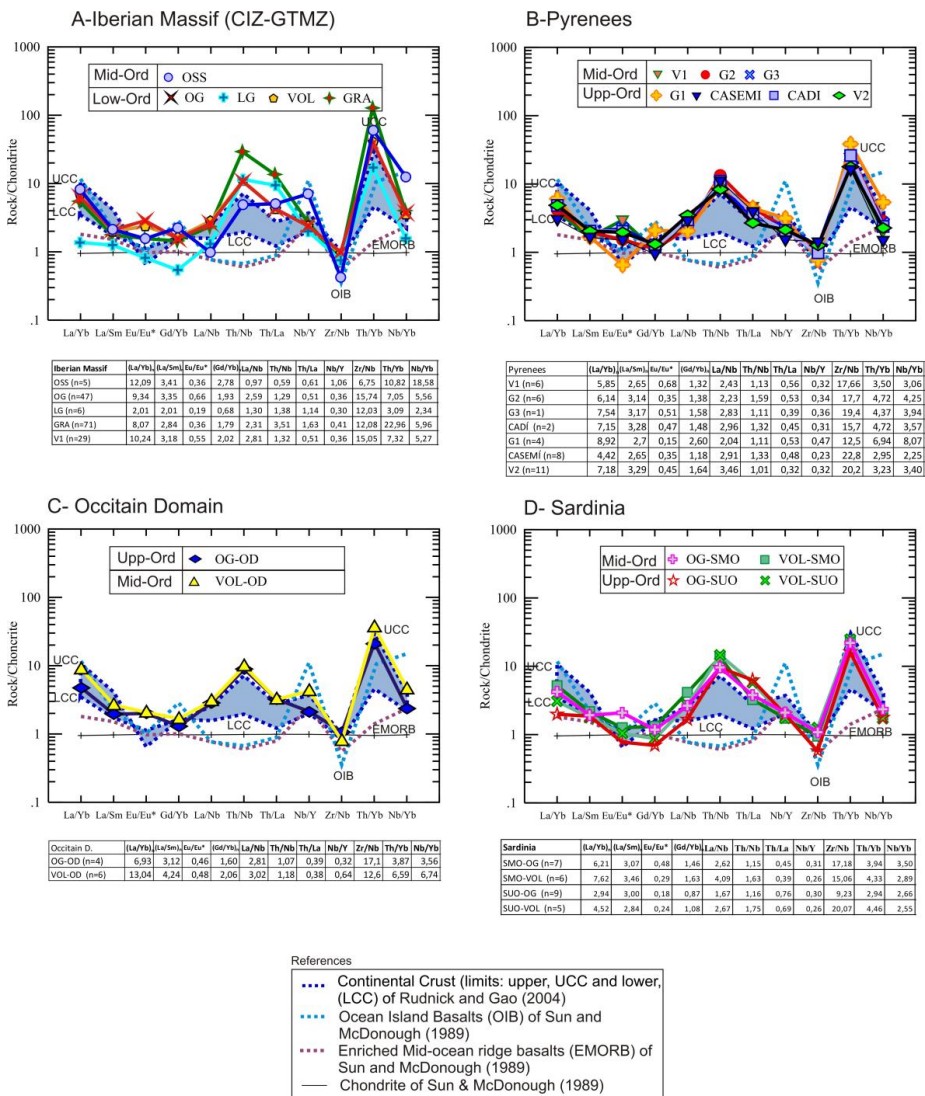

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

**Figure 8. Chondrite-normalised isotope ratio patterns (Sun and McDonough,**
**1989) for standard comparison for all study samples. Blue area: limits of**
**continental crustal values (Lower and Upper) of Rudnick and Gao (2003).**





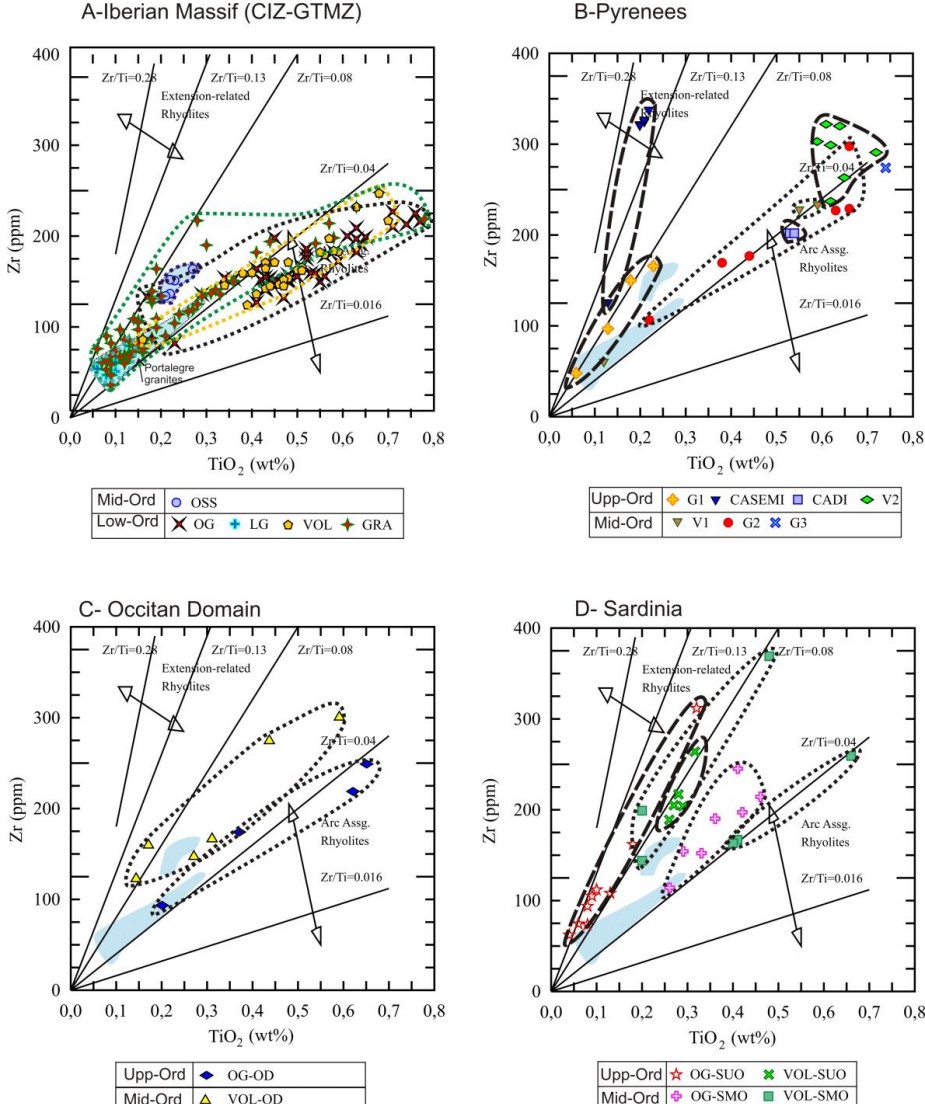

**Figure 9. Tectonic discriminating diagram of Zr vs. TiO$_2$ (Syme, 1998) for all study samples.**





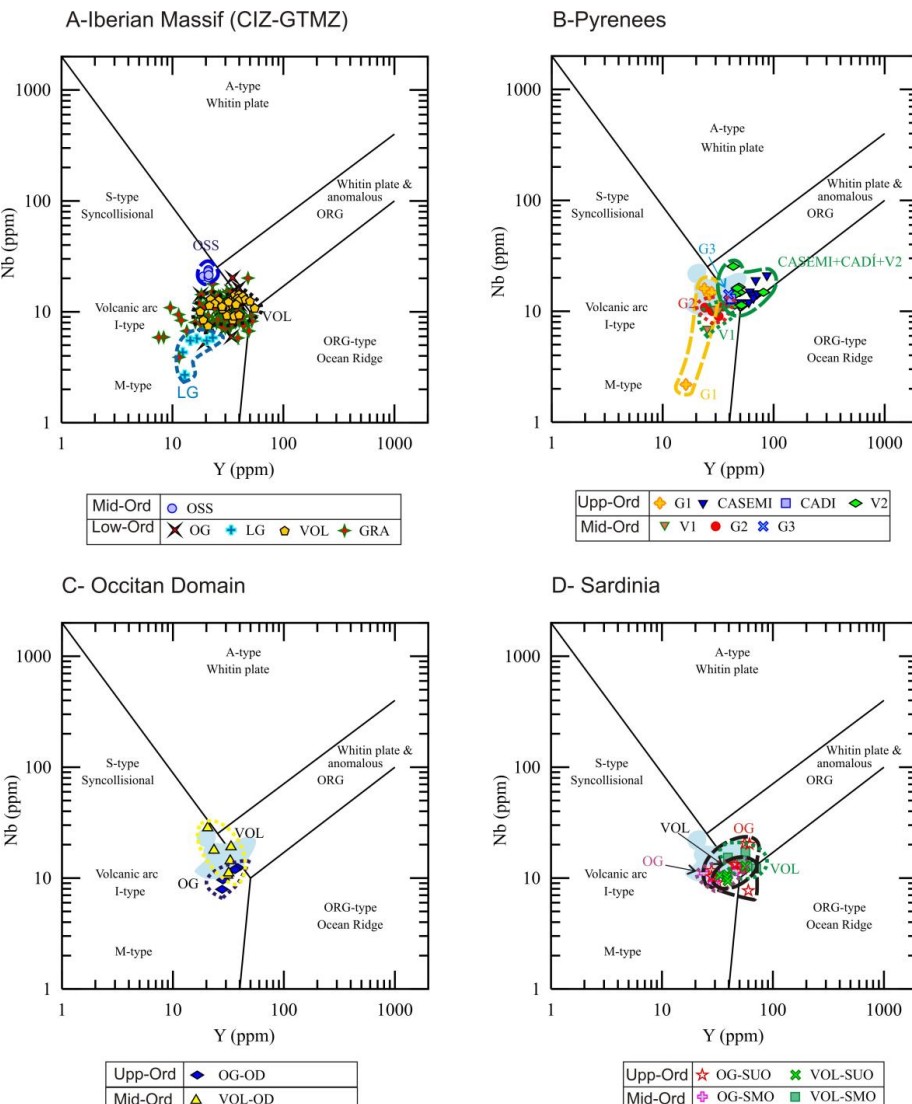

**Figure 10. Tectonic discriminating diagram of Y vs. Nb (Pearce et al., 1984) for all study samples.**





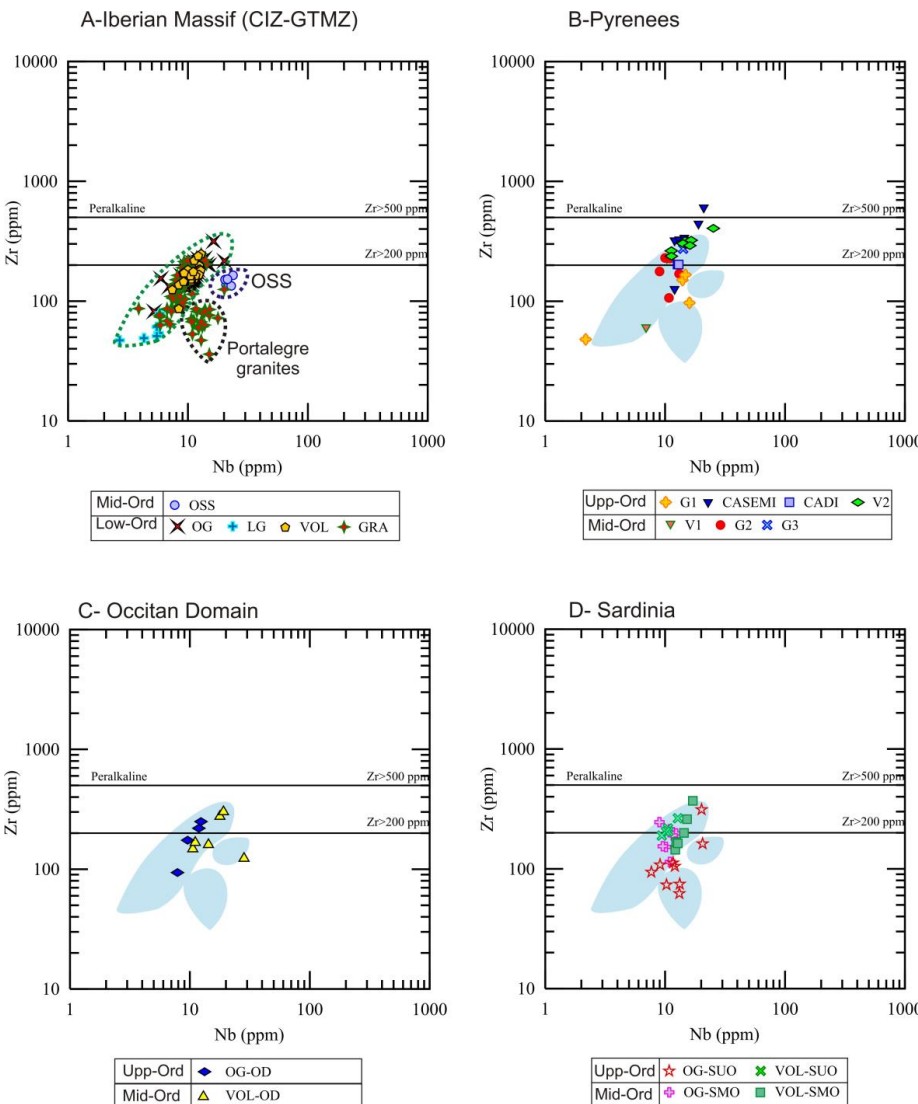


**Figure 11. Zr vs. 10$^4$ Ga/Al discrimination diagram (Whalen et al., 1987).**


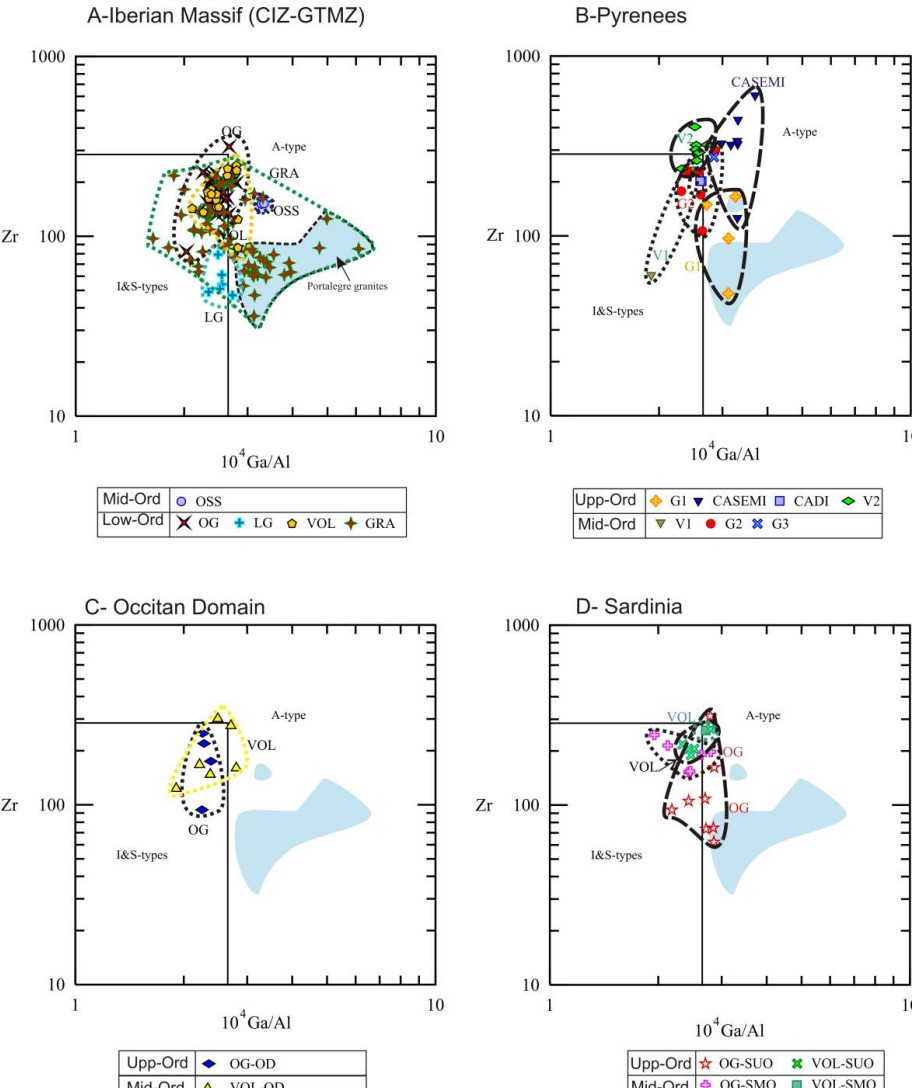


**Figure 12.** Zr–Nb plot diagram (Leat et al.,1986; modified by Piercey, 2011) for all

study samples.








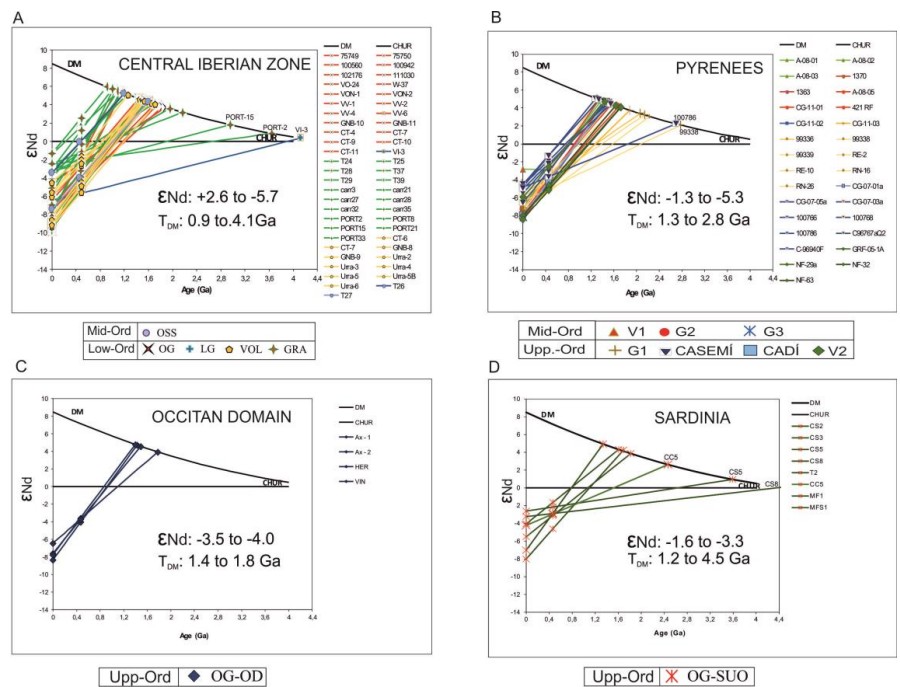

**Figure 13. εNd(t) vs. age diagram (DePaolo and Wasserburg, 1976; DePaolo, 1981) for study sampled. A. Central Iberian and Galicia-Trás-os-Montes Zones. B. Eastern Pyrenees. C. Occitan Domain. D. Sardinia; see references in the text.**





**TABLES**

| Sample | PYRENEES | | | MONTAGNE NOIRE | | | | SARDINIA | | | | | | | | Inner | Zone |
|---|---|---|---|---|---|---|---|---|---|---|---|---|---|---|---|---|---|
| | Albera | Pallaresa | Andorra | Axial | Zone | | | Externa | Zone | | | | | | | | |
| | A-08-03 | fC1803 | BN 1 | Ax - 1 | Ax - 2 | HER | VIN | CC 5 | CS 2 | CS 3 | CS 5 | CS 8 | MF 1 | MFS 1 | T 2 | PB50 | PB100 |
| $SiO_2$ | 68.38 | 71.67 | 69.18 | 70.38 | 67.43 | 68.31 | 73.97 | 76.43 | 75.14 | 76.52 | 76.61 | 76.36 | 72.13 | 75.94 | 75.55 | 68.93 | 67.24 |
| $TiO_2$ | 0.57 | 0.63 | 0.61 | 0.36 | 0.64 | 0.61 | 0.20 | 0.08 | 0.08 | 0.09 | 0.04 | 0.06 | 0.31 | 0.13 | 0.18 | 0.41 | 0.46 |
| $Al_2O_3$ | 15.68 | 14.24 | 15.05 | 14.90 | 15.76 | 15.39 | 13.82 | 13.28 | 12.81 | 11.80 | 12.71 | 12.63 | 13.80 | 13.16 | 12.94 | 16.32 | 15.79 |
| $Fe_2O_3$ | 4.09 | 4.54 | 4.20 | 3.04 | 4.11 | 4.19 | 2.05 | 0.69 | 1.39 | 1.44 | 1.28 | 1.35 | 2.96 | 1.55 | 1.62 | 3.19 | 4.78 |
| MnO | 0.07 | 0.06 | 0.05 | 0.04 | 0.04 | 0.04 | 0.04 | 0.01 | 0.01 | 0.01 | 0.01 | 0.01 | 0.02 | 0.03 | 0.04 | 0.08 | 0.08 |
| MgO | 1.35 | 0.78 | 1.16 | 0.78 | 1.33 | 1.34 | 0.43 | 0.08 | 0.15 | 0.16 | 0.06 | 0.05 | 0.36 | 0.19 | 0.08 | 1.15 | 1.58 |
| CaO | 0.21 | 0.53 | 1.78 | 1.22 | 1.44 | 1.58 | 0.62 | 0.32 | 0.25 | 0.15 | 0.20 | 0.35 | 0.61 | 0.38 | 0.17 | 3.05 | 2.70 |
| $Na_2O$ | 4.07 | 1.67 | 3.40 | 3.33 | 2.78 | 2.93 | 2.87 | 3.04 | 1.71 | 1.58 | 2.91 | 3.35 | 2.89 | 2.57 | 2.53 | 3.85 | 3.43 |
| $K_2O$ | 2.84 | 2.91 | 2.71 | 4.35 | 4.68 | 4.03 | 4.55 | 4.79 | 7.84 | 7.43 | 5.16 | 4.91 | 5.47 | 4.94 | 5.36 | 2.26 | 2.96 |
| $P_2O_5$ | 0.17 | 0.24 | 0.20 | 0.21 | 0.2 | 0.19 | 0.18 | 0.15 | 0.05 | 0.05 | 0.03 | 0.04 | 0.12 | 0.11 | 0.07 | 0.15 | 0.14 |
| L.O.I. | 2.03 | 2.60 | 1.50 | 1.2 | 1.3 | 1.2 | 1.2 | 1.1 | 0.4 | 0.7 | 0.9 | 0.8 | 1.1 | 0.9 | 1.4 | 0.90 | 0.70 |
| Total | 99.05 | 99.42 | 99.42 | 99.51 | 99.30 | 99.39 | 99.73 | 99.90 | 99.69 | 99.79 | 99.78 | 99.78 | 99.47 | 99.75 | 99.78 | 99.97 | 99.37 |
| | | | | | | | | | | | | | | | | | |
| As | 77.20 | 1.70 | 6.80 | 2.50 | 6.00 | 1.80 | 1.90 | 0.70 | 1.00 | 0.50 | 2.80 | 1.10 | 1.80 | 101.10 | 4.00 | 5.00 | 5.00 |
| Ba | 742.50 | 388.00 | 398.00 | 499 | 1050 | 767 | 256 | 60 | 467 | 109 | 21 | 27 | 784 | 194 | 192 | 689.00 | 600.00 |
| Be | 2.44 | 3.00 | 2.00 | 4.00 | 2.00 | 5.00 | 3.00 | 6.00 | 3.00 | 1.00 | 9.00 | 2.00 | 7.00 | 3.00 | 7.00 | 3.00 | 5.00 |
| Bi | 0.30 | 0.20 | 0.10 | 0.20 | 0.20 | 0.20 | 0.40 | 0.30 | 0.10 | 0.10 | 0.10 | 0.10 | 0.10 | 0.70 | 0.40 | 4.00 | 4.00 |
| Cd | 0.18 | 0.10 | 0.10 | 0.10 | 0.10 | 0.10 | 0.10 | 0.10 | 0.10 | 0.10 | 0.10 | 0.10 | 0.10 | 0.10 | 0.10 | | |
| Co | 5.84 | 4.60 | 6.20 | 5.20 | 5.20 | 5.40 | 2.70 | 0.50 | 1.60 | 1.00 | 0.80 | 0.60 | 2.30 | 1.50 | 1.20 | 5.00 | 14.00 |
| Cs | 9.79 | 5.60 | 4.90 | 14.30 | 7.10 | 6.80 | 7.30 | 4.20 | 3.40 | 1.60 | 4.50 | 4.60 | 6.40 | 3.90 | 4.10 | 4.20 | 9.40 |
| Cu | 16.34 | 13.20 | 10.30 | 7.20 | 7.40 | 10.10 | 8.70 | 4.70 | 4.60 | 8.20 | 26.80 | 2.50 | 5.00 | 5.50 | 5.00 | 10.00 | 60.00 |
| Ga | 21.03 | 19.80 | 18.80 | 19.10 | 19.20 | 18.90 | 16.70 | 19.30 | 14.90 | 15.30 | 19.40 | 19.20 | 20.70 | 19.00 | 19.90 | 17.00 | 18.00 |
| Hf | 6.40 | 7.30 | 6.40 | 5.00 | 6.90 | 5.70 | 3.10 | 3.10 | 4.10 | 4.30 | 3.50 | 3.80 | 8.80 | 3.70 | 5.80 | 5.90 | 5.30 |
| Mo | 1.20 | 0.90 | 1.00 | 0.60 | 0.90 | 0.60 | 0.30 | 0.70 | 0.70 | 0.70 | 0.80 | 0.50 | 1.70 | 0.80 | 1.60 | 2.00 | 2.00 |
| Nb | 10.49 | 11.30 | 11.30 | 9.60 | 12.40 | 11.90 | 7.90 | 10.30 | 7.70 | 12.10 | 13.20 | 13.30 | 20.20 | 9.10 | 20.60 | 9.00 | 11.00 |
| Ni | 16.56 | 8.00 | 7.70 | 20.00 | 20.00 | 20.00 | 20.00 | 20.00 | 20.00 | 20.00 | 20.00 | 20.00 | 20.00 | 20.00 | 20.00 | 20.00 | 80.00 |
| Pb | 7.94 | 9.80 | 22.90 | 3.50 | 4.60 | 5.10 | 3.60 | 2.90 | 7.40 | 8.60 | 4.50 | 5.50 | 5.10 | 6.30 | 5.50 | 21.00 | 24.00 |
| Rb | 124.40 | 123.70 | 137.20 | 204.6 | 161.6 | 142.2 | 188.2 | 289.9 | 206.1 | 187.4 | 294.1 | 275.1 | 208.7 | 256.4 | 227.1 | 85.00 | 118.00 |
| Sb | 2.27 | 0.10 | 0.30 | 0.10 | 0.10 | 0.10 | 0.10 | 0.10 | 0.10 | 0.10 | 0.10 | 0.10 | 0.10 | 0.10 | 0.10 | 5.00 | 5.00 |
| Sc | | 10.00 | 10.00 | 6.00 | 9.00 | 9.00 | 4.00 | 3.00 | 3.00 | 4.00 | 4.00 | 4.00 | 15.00 | 4.00 | 8.00 | 9.00 | 12.00 |
| Sn | 2.11 | 5.00 | 5.00 | 9.00 | 3.00 | 3.00 | 7.00 | 9.00 | 4.00 | 3.00 | 13.00 | 15.00 | 7.00 | 15.00 | 12.00 | 3.00 | 3.00 |
| Sr | 158.00 | 201.80 | 83.70 | 91.20 | 160.30 | 150.10 | 68.70 | 30.70 | 73.90 | 25.20 | 7.90 | 8.10 | 59.90 | 45.60 | 25.00 | 217.00 | 167.00 |
| Ta | 1.07 | 1.10 | 1.10 | 0.80 | 1.00 | 0.80 | 0.70 | 2.10 | 0.90 | 1.10 | 3.40 | 1.70 | 1.60 | 1.70 | 2.30 | 1.00 | 1.20 |
| Th | 11.90 | 15.70 | 13.50 | 11.10 | 14.40 | 14.30 | 5.90 | 9.10 | 14.10 | 17.00 | 13.50 | 13.10 | 22.80 | 10.20 | 26.90 | 13.30 | 11.50 |
| U | 3.70 | 5.10 | 4.60 | 4.10 | 3.60 | 3.20 | 4.80 | 3.30 | 2.90 | 3.20 | 3.50 | 3.50 | 4.60 | 8.10 | 4.90 | 4.50 | 2.20 |
| V | 44.49 | 49.00 | 36.00 | 36.00 | 63.00 | 68.00 | 22.00 | 8.00 | 8.00 | 8.00 | 8.00 | 8.00 | 15.00 | 8.00 | 10.00 | 62.00 | 53.00 |
| W | 1.80 | 1.90 | 2.50 | 3.20 | 2.60 | 1.60 | 3.00 | 5.60 | 0.90 | 2.10 | 5.20 | 3.00 | 2.40 | 4.40 | 3.50 | 1.00 | 20.00 |
| Y | 29.29 | 43.90 | 50.60 | 28.30 | 38.40 | 36.20 | 27.80 | 28.00 | 60.10 | 53.60 | 44.40 | 46.00 | 61.60 | 31.80 | 55.80 | 29.00 | 24.00 |
| Zn | 63.71 | 52.00 | 70.00 | 55.00 | 71.00 | 78.00 | 46.00 | 7.00 | 35.00 | 39.00 | 15.00 | 24.00 | 37.00 | 30.00 | 22.00 | 70.00 | 70.00 |
| Zr | 233.30 | 263.20 | 237.10 | 174.40 | 249.20 | 219.10 | 93.70 | 73.50 | 93.80 | #### | 62.20 | 74.50 | 311.80 | 108.10 | 161.90 | 245.00 | 214.00 |



| | | | | | | | | | | | | | | | | | |
|---|---|---|---|---|---|---|---|---|---|---|---|---|---|---|---|---|---|
| **La** | 27.90 | 45.30 | 38.00 | 29.60 | 39.50 | 38.70 | 13.60 | 10.50 | 22.70 | 19.50 | 12.10 | 13.40 | 54.20 | 17.90 | 31.30 | 26.90 | 34.30 |
| **Ce** | 59.00 | 86.90 | 75.50 | 58.10 | 77.00 | 78.20 | 26.70 | 21.60 | 42.10 | 39.70 | 26.20 | 29.90 | 109.80 | 37.40 | 97.60 | 53.20 | 70.50 |
| **Pr** | 7.26 | 9.80 | 8.47 | 6.99 | 9.41 | 9.55 | 3.36 | 2.36 | 4.73 | 4.85 | 3.00 | 3.24 | 11.94 | 4.07 | 6.86 | 5.88 | 8.20 |
| **Nd** | 27.83 | 35.60 | 31.20 | 26.00 | 36.40 | 36.40 | 12.60 | 8.40 | 16.60 | 17.10 | 10.50 | 10.90 | 44.70 | 15.00 | 24.00 | 21.60 | 29.40 |
| **Sm** | 5.80 | 7.69 | 7.16 | 5.70 | 7.55 | 7.63 | 3.15 | 2.43 | 4.10 | 4.41 | 3.28 | 3.44 | 9.37 | 3.88 | 4.93 | 4.70 | 6.00 |
| **Eu** | 0.98 | 1.05 | 1.03 | 0.87 | 1.27 | 1.15 | 0.41 | 0.14 | 0.43 | 0.13 | 0.06 | 0.09 | 1.17 | 0.30 | 0.19 | 0.95 | 0.93 |
| **Gd** | 5.22 | 8.32 | 7.89 | 5.59 | 7.28 | 7.05 | 3.38 | 3.20 | 5.60 | 5.50 | 4.42 | 4.69 | 10.60 | 4.50 | 6.34 | 4.00 | 5.10 |
| **Tb** | 0.87 | 1.26 | 1.27 | 0.89 | 1.17 | 1.10 | 0.67 | 0.69 | 1.13 | 1.18 | 1.03 | 1.07 | 1.70 | 0.82 | 1.27 | 0.70 | 0.80 |
| **Dy** | 5.30 | 6.68 | 8.00 | 5.09 | 6.89 | 6.39 | 4.59 | 4.30 | 7.69 | 8.23 | 7.31 | 7.66 | 10.28 | 5.24 | 9.00 | 3.70 | 4.30 |
| **Ho** | 1.06 | 1.52 | 1.73 | 0.99 | 1.42 | 1.30 | 0.98 | 0.91 | 1.91 | 1.91 | 1.59 | 1.65 | 2.13 | 1.12 | 2.01 | 0.70 | 0.80 |
| **Er** | 2.98 | 4.52 | 4.96 | 2.64 | 3.92 | 3.56 | 3.07 | 2.85 | 5.80 | 6.46 | 5.35 | 5.38 | 6.25 | 3.64 | 6.17 | 2.20 | 2.10 |
| **Tm** | 0.46 | 0.60 | 0.73 | 0.38 | 0.57 | 0.50 | 0.44 | 0.43 | 0.91 | 1.00 | 0.85 | 0.85 | 0.89 | 0.52 | 0.92 | 0.35 | 0.32 |
| **Yb** | 3.00 | 3.98 | 4.72 | 2.33 | 3.56 | 3.11 | 2.83 | 2.95 | 5.81 | 6.60 | 6.10 | 6.16 | 5.53 | 3.70 | 6.04 | 2.50 | 2.20 |
| **Lu** | 0.44 | 0.58 | 0.69 | 0.33 | 0.53 | 0.45 | 0.39 | 0.44 | 0.90 | 0.94 | 0.92 | 0.94 | 0.86 | 0.56 | 0.90 | 0.41 | 0.36 |

**Longitude** |°7'39.5063'' | °27'43.71'' |°33'29.3112'|°13'50.26''|°33'58.14''|2°57'58.80''E 2°13'50.21 °50'36.95'' °50'35.32'' °50'35.31'' °50'40.64'' °50'35.07''°50'46.57''|°52'01.84''|°48'54.23''E 9°09'32''E 9°09'32''E

**Latitude** |2°25'2.931'' |2°36'0.93'' |2°32'30.580|3°34'32.52'|3°29'3.27'|3°34'32.52'| 43°17'45.6)°54'15.91'3°52'38.36'3°52'38.37'°52'36.74'3°52'38.75'3°54'58.32'8°53'56.85''8°53'56.69'| 41°11'04''N 41°11'04''N

**Table 1.** Chemical analyses of magmatic rocks. ICP and ICP–MS methods at ACME–LABS in Canada.



| ZONES | SUBGROUPS | eNd$_{age}$ | | Tdm Ga) | | ($^{87}$Sr/$^{86}$Sr)$_{age}$ | |
|---|---|---|---|---|---|---|---|
| CIZ & | OG | -4,4 | | 1,58 | | 0,709 | |
| GTMZ | LG | -5,4 | -3,8 | 4,13 | 2,2 | 0,664 | 0,701 |
| | GRA | -1,6 | | 1,59 | | 0,698 | |
| | VOL | -3,6 | | 1,52 | | 0,732 | |
| | OSS | -2,0 | -2,0 | 1,40 | 1,4 | 0,711 | 0,711 |
| PYRENEES | V1 | -2,9 | | 1,36 | | | |
| | G2 | -3,8 | -3,6 | 1,49 | 1,5 | | |
| | G3 | -4,2 | | 1,50 | | | |
| | G1 | -4,2 | | 1,95 | | | |
| | CADÍ | -4,1 | -3,7 | 1,48 | 1,7 | | 0,701 |
| | CASEMÍ | -2,2 | | 1,61 | | 0,696 | |
| | V2 | -4,3 | | 1,63 | | 0,705 | |
| Occitan D. | OG-MOD | -3,9 | -3,9 | 1,52 | 1,5 | | |
| | VOL-MOD | | | | | | |
| SARDINIA | OG-MOS | | | | | | |
| | VOL-MOS | | | | | | |
| | OG-USO | -3,9 | -3,9 | 1,52 | 1,5 | | |
| | VOL-UOS | | | | | | |

**Table 2.** Average values of the different subgroups reported in the text.