# Peer review of "Comparative geochemical study on Furongian (Toledanian) and Ordovician (Sardic) felsic magmatic events in south-western Europe"

_Solid Earth, 2020_

## Referee Comment (RC1) · Laura Gaggero (Referee) · 21 May 2020

This manuscript addresses and brings clarity on the Early- Mid Ordovician felsic magmatism in different geological regions at the northern Gondwana margin and aims at comparing and conciliating the different petrogenetic models. The organization of the work is excellent, the literature review is very large and correct, and the overall goal of the manuscript is clear.

However, as I have more experience on Sardinia, among all zones, I observe that the comparison of chemical data was carried out on using the Mid and upper Ordovician data (Giacomini, Cruciani), while in the text it is clearly stated that also a Lower Or-

dovician magmatism is present in Sardinia. So, I suggest inserting in your comparison or quoting in the discussion also the bulk and isotopic data of Gaggero et al 2012 for the lower Ordovician felsic rocks, correctly cited in the text. The emphasis on the lower Ordovician magmatism from in Sardinia, that we ascribed to a magmatic starved incipient passive margin, can otherwise open to a link with the Toledanian phase in the Iberian Massif. In your model, Sardinia could represent a distal expression of the crustal melting after thermal doming.

I also bring to your attention the mid Ordovician andalusite thermal aureole around the Filau metagranites (Costamagna et al 2016, Lithos) that constrains the emplacement level of the felsic rocks.

Finally : monacite at line 300. Congratulation for your interesting work, and kindest regards.

Laura Gaggero

---

## Referee Comment (RC2) · Jochen Mezger (Referee) · 26 May 2020

General comments: This study compares the geochemistry of two distinct igneous felsic magmatic events in southwestern Europe during the Ordovician, which marked the northern Gondwana at that time. These magmatic events took place when compressive tectonics were absent and are generally associated with rifting of the continental margin related to the opening of the Rheic Ocean. What makes this study so interesting is that it is the first one to compare a large number of published geochemical data of igneous rocks with known emplacement ages in order to find out if common geodynamic settings can be attributed to these magmatic pulses. Although some new

geochemical data is presented, it is mainly a review paper. The main problem is that a lot of data is being presented and discussed without providing a sufficient overview, leaving the reader a bit lost. A more concise presentation of the published data, and a discussion how their new analytical data adds to the understanding of the geodynamic setting would improve the paper. In the discussion of the geodynamic setting of the Ordovician magmatic events it is not always clear what is a recapitulation of other authors and their own contribution. Better structuring should make this more clear.

Specific comments: The title is too general and unimaginative, suggesting that the paper only presents data. The key finding of this study should be reflected in the title. If I am correct, the Toledanian phase lasted into the early Ordovician. If so then the title is misleading as it reads ". . .Furongian (Toledanian) and Ordovician (Sardic) felsic magmatic events. . ." The introduction could be improved by stating the problem and the objective of the study, the latter of which is listed in the final paragraph. Also, it would be helpful to give an approximate time frame of the Toledanian and Sardic phases. Some statements in the first paragraph ("but they are related to neither metamorphism nor penetrative deformation", line 57) should be accompanied with key references. The author's own new analytical data should also be mentioned in the introduction with a justification on why it was deemed necessary. As it is, there is no mention of it and the reader has the impression that this is purely a review paper. Geologic Setting: A lot of geochronological data is presented with detailed listing of the age uncertainties, e.g. 478.1 +/- 1.2 Ma. Since it is not their own data, this can be represented as ca. 478 Ma. And instead of listing every single age of an orthogneiss complex, the ages of a zone can be summarized, e.g. 471-450 Ma for the migmatitic orthogneisses of the Montagne Noire (lines 289-291). When the authors discuss the Pyrenees, they refer to the Eastern Pyrenees. While most of the data is from the Eastern Pyrenees, the Aston and Hospitalet domes, discussed by Denèle et al. and Mezger & Gerdes, are located in the Central Pyrenees. So, I would refer to chapter 2.2 as "Central and Eastern Pyrenees". In line 279 they refer to "augen gneisses" (the actual spelling is "augengneiss) as metamorphic high-grade gneisses. I don't think that is correct. The term augengneiss refers

to the microstructure, large augen (commonly, but not restricted to K-feldspar) in a finer grained matrix, mainly in metagranites. There is no direct metamorphic association, although most metagranitic augengneisses are probably amphibolite facies. At last, a map showing the trend of Ordovician ages throughout western Europe would nicely summarize this chapter and provide some needed overview.

Geochemical data: Since the authors also present new data, a paragraph on the analytical methodology should be included, as well as where the analyses were made. This is completely missing. Similar to the Geologic Setting chapter, a lot of detailed geochemical data is presented, making for a repetitive reading. Most of the major elements data can be represented in an extra figure, and individual magmatic suites referred to as "potassium-rich dacite to rhyolite" (line 417) without listing the range of major elements. The discussion of epsilon Nd data is a bit spotty. First, it is unclear in the text what epsilon Nd values are discussed (line 422). Obviously, they are not the present day values but those at the time of emplacement. Second, line 429 refers to erroneous TDM values, without elaborating what they are. Third, in the same sentence, a 147Sm/144Nd ratio of greater than 0.13 is considered high. That is an average value even for felsic rocks, mafic and ultramafic rocks can have ratios of 0.3. There needs to be some clarification.

Interpretation of epsilon Nd values: The second last paragraph (lines 730-733) states that very little variation in epsilon Nd values is a sign of magmas derived from young crustal rocks. An epsilon Nd value per se does not indicate the age of a rock, but rather how much the protolith melt was evolved. Negative epsilon Nd values of -3.5 to -4.0 indicate moderately evolved protoliths, not an Archean continental margin, but also not a juvenile volcanic arc. Likewise, referring to depleted mantle model ages of 1.8 to 1.4 Ga do not reflect a short crustal residence time. To summarize, the discussion and interpretation of Nd data requires some revision.

Discussion: The geographic trend of younger ages of Ordovician magmatism is not discussed. Is there a link between the Toledanian and Sardic phases or are these strictly bounded to regions, CIZ and Pyrenees and north thereof, respectively?

Technical corrections: Here I mainly refer to the figures and tables. Typos and minor grammatical errors are flagged in the annotated PDF that is attached to this review. Fig. 1: The sample numbers are very hard to read. Even when considering that figures can be viewed enlarged online. The majority of the sample localities in 1B are not discussed in the paper. So why listing them all in the figure captions? For easier location of the individual regions, add the region name to 1B through E. Figs. 2, 8 and 13: the labels are much too small. Fig. 5: Place symbols as inset in the figure instead of referring to the legend of a previous figure. Fig. 9: What do the double-sided arrows signify? Table 1: Add a vertical line separating the different regions to enhance orientation. Information on the lab that did the analyses should be included in a footnote or the table caption. The sample location (lat/long) should be moved to the column header. Latitudinal and longitudinal data are listed up to the fourth decimal of a second! Just as a reminder, one second latitude represents approximately 30 m. It is more than sufficient to report full seconds. Table 2: It consists only of already published data. This is not evident from the table caption. The table shows several rows without any data. Is there a purpose? Sr isotope data are listed in the table, but they are not discussed in the text. Why? If not necessary, that data should he deleted.

Please also note the supplement to this comment:
https://www.solid-earth-discuss.net/se-2020-45/se-2020-45-RC2-supplement.pdf

**Supplement:**

[Figure]

[Figure]

[revised manuscript text omitted]
** l°7'39.5063'' l°27'43.71'' l°33'29.3112'l°13'50.26''l°33'58.14''l2°57'58.80''E 2°13'50.21 °50'36.95'' l°50'35.32'' l°50'35.31'' l°50'40.64'' l°50'35.07'' l°50'46.57'' l°52'01.84''l3°48'54.23''E 9°09'32''E 9°09'32''E

**Latitude** l2°25'2.931'' ll2°36'0.93'' l2°32'30.580l3°34'32.52''l3°29'3.27''l3°34'32.52'' l43°17'45.6l°54'15.91'3°52'38.36'3°52'38.37l°52'36.74'3°52'38.75'3°54'58.32'8°53'56.85''l8°53'56.69''l41°11'04''N 41°11'04''N

**Table 1.** Chemical analyses of magmatic rocks. ICP and ICP–MS methods at ACME–LABS in Canada.

[Figure]

[Figure]

| ZONES | SUBGROUPS | eNd$_{age}$ | | Tdm Ga) | | ($^{87}$Sr/$^{86}$Sr)$_{age}$ | |
|---|---|---|---|---|---|---|---|
| CIZ & | OG | -4,4 | | 1,58 | | 0,709 | |
| GTMZ | LG | -5,4 | -3,8 | 4,13 | 2,2 | 0,664 | 0,701 |
| | GRA | -1,6 | | 1,59 | | 0,698 | |
| | VOL | -3,6 | | 1,52 | | 0,732 | |
| | OSS | -2,0 | -2,0 | 1,40 | 1,4 | 0,711 | 0,711 |
| | V1 | -2,9 | | 1,36 | | | |
| | G2 | -3,8 | -3,6 | 1,49 | 1,5 | | |
| | G3 | -4,2 | | 1,50 | | | |
| PYRENEES | G1 | -4,2 | | 1,95 | | | |
| | CADÍ | -4,1 | -3,7 | 1,48 | 1,7 | | 0,701 |
| | CASEMÍ | -2,2 | | 1,61 | | 0,696 | |
| | V2 | -4,3 | | 1,63 | | 0,705 | |
| Occitan D. | OG-MOD | -3,9 | -3,9 | 1,52 | 1,5 | | |
| | VOL-MOD | | | | | | |
| | OG-MOS | | | | | | |
| SARDINIA | VOL-MOS | | | | | | |
| | OG-USO | -3,9 | -3,9 | 1,52 | 1,5 | | |
| | VOL-UOS | | | | | | |

**Table 2.** Average values of the different subgroups reported in the text.

---

## Author Comment (AC1) · 26 Jun 2020

REVIEWER – LAURA GAGGERO

Thanks for your revision and constructive remarks. Regarding your comments:

I suggest inserting in your comparison or quoting in the discussion also the bulk and isotopic data of Gaggero et al. (2012) for the lower Ordovician felsic rocks, correctly cited in the text. The emphasis on the lower Ordovician magmatism from in Sardinia, which we ascribed to a magmatic starved incipient passive margin, can otherwise open to a link with the Toledanian phase in the Iberian Massif. In your model, Sardinia could

represent a distal expression of the crustal melting after thermal doming.

Thanks for sending us (JMC) the dataset. We have studied it in detail and found some inconveniences for including these geochemical values. Many of the Furongian-Lower Ordovician samples of Gaggero et al. (2012) display alkaline to subalkaline affinities. Alkaline ORD19 is a probable Furongian sample; and alkaline ORD45 and 47 samples come from the Li Trumbetti Unit (Inner Zone), and these allochthonous zones are not considered in our paper. In addition, subalkaline ORD25 and 34 samples are andesites-to-basaltic andesites, so basic rocks, and the topic of the paper is the geochemical comparison of felsic/acidic rock samples. In addition, ORD34 was also sampled from the Vanaglia Unit, in the allochthonous Inner Zone, so beyond the study area. However, we have included in the second version the isotopes of sample OD31, which fulfil the requirements of the paper, including felsic samples from the Furongian-Ordovician of the Outer Zones of Sardinia.

I also bring to your attention the mid Ordovician andalusite thermal aureole around the Filau metagranites (Costamagna et al 2016, Lithos) that constrains the emplacement level of the felsic rocks. Finally : monacite at line 300.

We have added the metamorphic conditions of the Filau metagranites and properly written "monacite". Thanks again.

---

## Author Comment (AC2) · 26 Jun 2020

REVIEWER – JOACHEM MEZGER

Many thanks for such a detailed revision. All the words and phrases highlighted in the Similarity Report have been re-written following your advice, except one question: "upper Cambrian" is a former statement of Furongian (we indicate now both terms).

The main problem is that a lot of data is being presented and discussed without providing a sufficient overview, leaving the reader a bit lost. A more concise presentation of the published data, and a discussion how their new analytical data adds to the understanding of the geodynamic setting would improve the paper. In the discussion of the geodynamic setting of the Ordovician magmatic events it is not always clear what is a recapitulation of other authors and their own contribution. Better structuring should make this more clear.

The overview is clearly established in the figures 1 (tectonostratigraphic setting of samples), 2 (litho- and chronostratigraphic emplacement of samples) and Table 2 (summary of the geochemical features shown by the distinguished magmatic groups in which all the dataset has been subdivided). In the Introduction, it is now highlighted that "The re-appraisal is based on 17 new samples from the Pyrenees, Montagne Noire and Sardinia, completing the absence of analysis in these areas and wide-ranging a dataset of 93 previously published geochemical analyses throughout the study region in south-western Europe." When the paper documents the geochemical subdivision of the study samples (new and previously published), we have differentiated new and already known data. However, in the geochemical discussion, we have characterized and discussed the geochemical dataset with its subdividing groups. As stated above, the new data allow completing the precious incomplete dataset from Montagne Noire and Sardinia.

Specific comments: The title is too general and unimaginative, suggesting that the paper only presents data. The key finding of this study should be reflected in the title.

We follow referee's advice and added a second subtitle explaining the geodynamic significance of the paper: "underplating of hot mafic magmas linked to the opening of the Rheic Ocean".

If I am correct, the Toledanian phase lasted into the early Ordovician. If so then the title is misleading as it reads ". . .Furongian (Toledanian) and Ordovician (Sardic) felsic magmatic events. . ."

Yes, we have adapted the title updating the "Furongian–earliest Ordovician (Toledanian)" character.

The introduction could be improved by stating the problem and the objective of the study, the latter of which is listed in the final paragraph. Also, it would be helpful to give an approximate time frame of the Toledanian and Sardic phases.

The problem was (and is) stated in the last paragraph: "Until now the Toledanian and Sardic magmatic events had been studied on different areas and interpreted separately, without taking into account their similarities and differences. In this work, the geochemical affinities of the Furongian-Early Ordovician (Toledanian) and Early-Late Ordovician (Sardic) felsic magmatic activities recorded in the Central Iberian and Galicia-Trás-os-Montes Zones, Pyrenees, Occitan Domain and Sardinia are compared." The main purpose is written in the last sentence: "This comparison may contribute to a better understanding of the meaning and origin of this felsic magmatism, and thus, to discuss the geodynamic scenario of this Gondwana margin during Cambrian–Ordovician times". The time frame of the Toledanian and Sardic Phases is repeated several times in the Introduction: "Furongian–Early Ordovician (Toledanian) and Early–Late Ordovician (Sardic)". Later the referee suggests adding a new figure with a chronological distribution of the Toledanian and Sardic magmatic activities: this is now included in the new figure 3 (Relative probability plots of the age of the Cambrian–Ordovician magmatism).

Some statements in the first paragraph ("but they are related to neither metamorphism nor penetrative deformation", line 57) should be accompanied with key references.

Done.

The author's own new analytical data should also be mentioned in the introduction with a justification on why it was deemed necessary. As it is, there is no mention of it and the reader has the impression that this is purely a review paper.

We repeat the response to the first query: "The re-appraisal is based on 17 new samples from the Pyrenees, Montagne Noire and Sardinia, completing the absence of analysis in these areas and wide-ranging a dataset of 93 previously published geochemical

analyses throughout the study region in south-western Europe."

Geologic Setting: A lot of geochronological data is presented with detailed listing of the age uncertainties, e.g. 478.1 +/- 1.2 Ma. Since it is not their own data, this can be represented as ca. 478 Ma. And instead of listing every single age of an orthogneiss complex, the ages of a zone can be summarized, e.g. 471-450 Ma for the migmatitic orthogneisses of the Montagne Noire (lines 289-291).

We follow this advice and have maintained the age uncertainties only in the figure captions.

When the authors discuss the Pyrenees, they refer to the Eastern Pyrenees. While most of the data is from the Eastern Pyrenees, the Aston and Hospitalet domes, discussed by Denèle et al. and Mezger & Gerdes, are located in the Central Pyrenees. So, I would refer to chapter 2.2 as "Central and Eastern Pyrenees".

It is true. We have updated the text to the description and interpretation of samples from the Central and Eastern Pyrenees.

In line 279 they refer to "augen gneisses" (the actual spelling is "augengneiss) as metamorphic high-grade gneisses. I don't think that is correct. The term augengneiss refers C2 SED Interactive comment Printer-friendly version Discussion paper to the microstructure, large augen (commonly, but not restricted to K-feldspar) in a finer grained matrix, mainly in metagranites. There is no direct metamorphic association, although most metagranitic augengneisses are probably amphibolite facies.

We explain now this point by explaining the equivalence between "augen gneiss" and "augengneiss" (lines 286-287).

At last, a map showing the trend of Ordovician ages throughout western Europe would nicely summarize this chapter and provide some needed overview.

As explained above, this has been done in the new figure 3.

Geochemical data: Since the authors also present new data, a paragraph on the analytical methodology should be included, as well as where the analyses were made. This is completely missing.

A new section entitled "Material and methods" is added explaining the analytical methodology and the labs where the geochemical analyses were made.

Similar to the Geologic Setting chapter, a lot of detailed geochemical data is presented, making for a repetitive reading. Most of the major elements data can be represented in an extra figure, and individual magmatic suites referred to as "potassium-rich dacite to rhyolite" (line 417) without listing the range of major elements.

We have deleted the descriptive repetitions made in this section and summarized them in the new Table 2.

The discussion of epsilon Nd data is a bit spotty. First, it is unclear in the text what epsilon Nd values are discussed (line 422). Obviously, they are not the present day values but those at the time of emplacement.

We refer to isotopic ÆŘNd(t) values: the suffix "(t)" is added throughout the paper.

Second, line 429 refers to erroneous TDM values, without elaborating what they are. Third, in the same sentence, a 147Sm/144Nd ratio of greater than 0.13 is considered high. That is an average value even for felsic rocks, mafic and ultramafic rocks can have ratios of 0.3. There needs to be some clarification.

Yes, this point was not discussed in detail. We have explained (see lines 644-654) that: "display anomalous TDM values and 147Sm/144Nd ratios > 0.17 (Table 2; Fig. 14), a character relatively common in some felsic rocks (DePaolo, 1988; Martínez et al., 2011). According to Stern et al. (2012), these values should not be considered, but a possible explanation for these high ratios may be related to the M-type tetrad effect (e.g., Irber, 1999; Monecke et al., 2007; Ibrahim et al., 2015), which affects REE fractionation in highly evolved felsic rocks due to the interaction with hydrothermal fluids.
This process can be reflected as an enrichment of Sm related to Nd. Other authors, however, explain this enrichment as a result of both magmatic evolution (e.g., McLennan, 1994; Pan, 1997) and weathering processes after exhumation (e.g., Masuda and Akagi, 1989; Takahasi et al., 2002)".

Interpretation of epsilon Nd values: The second last paragraph (lines 730-733) states that very little variation in epsilon Nd values is a sign of magmas derived from young crustal rocks. An epsilon Nd value per se does not indicate the age of a rock, but rather how much the protolith melt was evolved. Negative epsilon Nd values of -3.5 to -4.0 indicate moderately evolved protoliths, not an Archean continental margin, but also not a juvenile volcanic arc. Likewise, referring to depleted mantle model ages of 1.8 to 1.4 Ga do not reflect a short crustal residence time. To summarize, the discussion and interpretation of Nd data requires some revision.

This point has been revised and developed in the new version based on the M-type tetrad effect (e.g., Irber, 1999; Monecke et al., 2007; Ibrahim et al., 2015).

Discussion: The geographic trend of younger ages of Ordovician magmatism is not discussed. Is there a link between the Toledanian and Sardic phases or are these strictly bounded to regions, CIZ and Pyrenees and north thereof, respectively?

Hope this point has been solved by including the new figure 3.

Technical corrections: Here I mainly refer to the figures and tables. Typos and minor grammatical errors are flagged in the annotated PDF that is attached to this review. Fig. 1: The sample numbers are very hard to read. Even when considering that figures can be viewed enlarged online. The majority of the sample localities in 1B are not discussed in the paper. So why listing them all in the figure captions? For easier location of the individual regions, add the region name to 1B through E. Figs. 2, 8 and 13: the labels are much too small. Fig. 5: Place symbols as inset in the figure instead of referring to the legend of a previous figure. Fig. 9: What do the double-sided arrows signify? Table 1: Add a vertical line separating the different regions to

enhance orientation. Information on the lab that did the analyses should be included in a footnote or the table caption. The sample location (lat/long) should be moved to the column header. Latitudinal and longitudinal data are listed up to the fourth decimal of a second! Just as a reminder, one second latitude represents approximately 30 m. It is more than sufficient to report full seconds. Table 2: It consists only of already published data. This is not evident from the table caption. The table shows several rows without any data. Is there a purpose? Sr isotope data are listed in the table, but they are not discussed in the text. Why? If not necessary, that data should he deleted.

Following the journal's rules, all the tipos and labels are greater than "Arial 7 pt". Only the localities reported in the text are now referred to in figure 1. In some cases, the figures are so complex that we have explained the symbols outside the figures; otherwise, the result was unreadable. The double-arrows of figure 9 (now 10) were used in Syme's (1998) original definition; now, they are explained in the figure caption. Table is now better arranged and their latitude/longitude data are documented with a single decimal. Table 2 is completely updated summarizing the geochemical data reported in the text (Sr isotopic data are, of course, deleted).

---

## Editor Decision (ED1)

[revised manuscript text omitted]

References

[revised manuscript text omitted]

---

## Author Response (AR2)

**To Topical Editor:** Juan Gómez-Barreiro

Dear editor,

Thanks for your message. It is rather unusual to receive a third revision from an editor, but we have tried to follow your last remarks. See below some explanations, because we have not properly understood some of your queries.

*A) It is not clear how the authors choose the studied areas in the Iberian Massif. A short description on these criteria could be very useful for the interested reader.*

The criteria for the selection of the targeted study areas was (and still is) explained at the end of the Introduction section. "Until now the Toledanian and Sardic magmatic events had been studied on different areas and interpreted separately, without taking into account their similarities and differences. In this work, the geochemical affinities of the Furongian–Early Ordovician (Toledanian) and Early–Late Ordovician (Sardic) felsic magmatic activities recorded in the Central Iberian and Galicia-Trás-os-Montes Zones, Pyrenees, Occitan Domain and Sardinia are compared. The re-appraisal is based on 17 new samples from the Pyrenees, Montagne Noire and Sardinia, completing the absence of analysis in these areas and wide-ranging a dataset of 93 previously published geochemical analyses throughout the study region in south-western Europe".

• *Besides, according to up-to-date references (e.g. Martínez Catalán et al 2019; https://doi.org/10.1007/978-3-030-10519-8_4), the Cantabrian, Westasturian-Leonese and Central Iberian zones were part of the Gondwana margin at that time span (broadly autochthon), while in the Galicia -Trás-os-Montes zone (Allochthon), only those units below the Ophiolites are clearly of that affinity (Basal and Parautochthon units). This connects with the non-usual division of the Iberian Massif in the Figure 1 which is not explained in the text. Please better use para-autochthon from the greek Παρά.*

We agree in considering the CZ, WALZ and CIZ as part of the Gondwana margin. However, these units cannot be considered as a Variscan "autochthon" as they are overthrusting them toward the (present-day) NE. So the question is: do we use the term "autochthon s.l.", the term "(par-)autochthon" (as suggested in the previous version) or simply "autochthon" (as many colleagues suggest though all of us consider this term as a very broad approximation)? A parautochthon is always "by comparison with" another unit, so this term fits well for the CZ, WALZ and CIZ units. We know that the "official terminology" used by Variscan-ologists is to call this "Autochthon" but we politely disagree.
We do not understand which is the "non-usual division" of the Iberian Massif used in Fig. 1 (probably you refer to Fig. 1A). The previous works on which this figure is based occur in the figure caption. We have modified the outline of the GTMZ in Figs. 1A and. 1B; maybe this was what you suggested in your query.

• *Some suggestions about other parts of the Figure 1 (e.g. 82-Saldanha and 90-Urra are not in the map!) are included as comments in the PDF attached to this report.*

The tiny outcrops of 82-Saldanha and 99 (now 90)-Urra are indicated in the figure.

B). *Regarding the geological setting it is important to note that:*
*B. 1) In the Central Iberian Zone an Upper Ordovician unconformity has been identified in two zones: the Truchas syncline and to the east of the Morais Allochthonous Complex. Data could be raised from Martínez Catalan et al. (1992) and Sarmiento et al. (1999) for the earlier, and from Dias da Silva et al. (2011), Dias da Silva et al. (2014) and Dias da Silva et al. (2016) for the latter. This data may well be included in the paper for their relevance, either for stating the existence of this unconformity, or for refuting it based on later/own data.*

We disagree. It is obvious we have different perceptions of the same data.
(i) In the Truchas Syncline, Martínez Catalán et al. (1992) suggested the presence of some "syn-sedimentary normal faults, though they had probably some strike-slip component, and

gave rise to a half-graben in which a syn-rift sequence was deposited", based on distinct modifications in the thickness of the involved formations.

In contrast, Sarmiento et al. (1999) clearly indicated that a conodont-based biostratigraphic analysis contradicts a part of the synsedimentary tectonic model proposed by Martínez Catalán et al. (1992). The former authors described a Hirnantian glaciogenic unconformity capping a conformable succession formed by the Luarca and Casaio formations. In fact, the authors stated that "there are two opposed interpretations for the Upper Ordovician sedimentary context in the area, one based on the Hirnantian glaciogenic activity" (Sarmiento et al., 1999) and the other on extensional tectonic processes (Martínez Catalán et al., 1992). The latter was explained as a result of rift-related extensional pulses leading to the record of horst-and-graben palaeotopographies sealed by overlying strata (re-dated by Sarmiento et al. as Hirnantian and glaciogenic in character).

In any case, to avoid everlasting discussions, this "apparent" unconformity (or unconformities if they are several ones) could be related either to (i) glaciogenic activity (the chronostratigraphic conodont-based control of the Rozadais Formation precludes the interpretation of its base as tectonically induced; and (ii) extensional pulses leading to the onset of (half-)grabens. Even accepting the second interpretation, these events are overabundant in the Cambrian rifting and Ordovician passive-margin framework of SW Europe. These discontinuities are not marked in the figure 2 because they are abundant, such as those marking the lower-middle Cambrian transition in the CZ, the Iberian Chains and the Montagne Noire. We have not included them and we should not make an exception for the Truchas syncline. An unconformity is not necessarily equivalent to a tectono-thermal event; on the contrary, many of them can be interpreted as a result of sea-level changes, and these are not issued in our paper.

(ii) In the Morais Allochthonous Complex, an interpretative figure occurs in Dias da Silva (2013), interpreting this unconformity as a result of an extensional pulse related to a low-angle discordance and the distal disappearance of the Moncorvo black shales. Again, this is a syn-sedimentary unconformity associated with extensional conditions, associated with basic volcanics (basic not felsic, so no considered in our work) and similar to many other local extensional breakdowns that are not considered in our work. Therefore, we should not include them.  Please, do not tell us that you consider this event as representative of the Sardic Phase simply because it is written in some papers. No arguments support this proposal.

[Figure]

Figura 3.31 - Esquema interpretativo de la disposición de las unidades vulcano-carbonatadas de la Formación Santo Adrião. A semejanza del esquema propuesto para la Caliza de la Aquiana y la Formación Agüeira al Norte del Sinforme de Truchas (Martínez Catalán *et al.*, 1990), la sedimentación detrítica se desarrolla en lugares marginales a las plataformas carbonatadas, donde las condiciones no son ideales para la formación de estructuras biohérmicas. La erosión previa al depósito del complejo vulcano-carbonatado produjo la discordancia angular con las formaciones de Marão y Vale de Bojas y la total desaparición del Ordovícico medio por debajo de la Formación Santo Adrião.

Let's read before what Da Silva et al. (2016) have written. They consider this discontinuity as the Sardic unconformity because: "The regressive character of the basal Upper Ordovician sedimentary record in both domains points to the formation of horsts and half-grabens of local extent combined with the deposition of limestone beds in the shallower areas, deposited over the Sardic Unconformity" (Fig. 3). It reads in Fig. 12 caption: "Tilting and gentle folding of the Lower-Middle Ordovician strata, due to the rotation of individual half-grabens and horsts, create the Sardic Unconformity in Iberia". So, it is clear that Da Silva and co-authors are proposing a Sardic Phase marking the base of the Upper Ordovician in the GTMZ and the CIZ simply based on the record of extensional breakdowns affecting the basement and developing normal faults. This is not at all the Sardic Phase, but a localized extensional pulse with no cortical uplift + erosion of the uplifted areas under subaerial exposure + intrusion of calc-alkaline granites (now preserved as orthogneisses) + record of gaps of about 25-30 m.y. + record of alluvial-fluvial deposits onlapping the unconformity.

Although we would have preferred avoiding entering in this discussion, your remark has encouraged us to add a new paragraph discussing Días da Silva et al.'s surprising idea:

"The Sardic Phase has been somewhat proposed marking a stratigraphic discontinuity close to the Middle-Upper Ordovician boundary interval in some areas of the Central Iberian (e.g., Buçaco and the Truchas Syncline; Martínez Catalán et al., 1992) and the Morais Allochthonous Complex of the Galicia-Trás-os-Montes Zones (Días da Silva et al., 2011, 2014, 2016; Días da Silva, 2013). In the Truchas Syncline, the significance of the discontinuity (or discontinuities) was questioned by a biostratigraphic study of conodonts and the re-interpretation of some of these scouring surfaces as the result of Hirnantian glaciogenic incisions (Sarmiento et al., 1999). The pre–Hirnantian discontinuities have been interpreted as linked to the development of "horsts and half-grabens of local extent" , as a result of which "tilting and gentle folding of the Lower–Middle Ordovician strata, due to the rotation of individual half-grabens and horsts, create the Sardic unconformity in Iberia" (Da Silva et al., 2016: pp. 1131 and 1143). However, the presence of synsedimentary listric faults associated with local outpouring of a basic volcanism, related to extensional pulses in the Ordovician passive-margin platform fringing Northwest Gondwana, cannot be associated with the Sardic Phase. As summarized in this work, the Sardic Phase is characterized by generalized cortical uplift, denudation of exposed uplifted areas under subaerial exposure, stratigraphic gaps of about 25–30 m.y., broad intrusion of felsic granitic plutons (now orthogneisses after Variscan deformation and metamorphism) with calc-alkaline affinity, and record of alluvial-to-fluvial deposits onlapping the unconformity. These are the features that characterize the Ordovician Sardic Phase, not the record of Ordovician volcanism and of local listric faults. In contrast. the Sardic aftermath is represented by abundant basic volcanic activity, mainly of tholeiitic  affinity, and lining rifting branches highlighting the onset of listric-fault networks; this event could be compared with some processes recorded in the Central Iberian and the Galician-Trás-os-Montes Zones, but not with the Sardic Phase. Therefore, the presence of the Sardic Phase in Iberia was already ruled out by the information published during the last two decades, and should not be maintained except if the above-reported tectonothermal events are really found in Iberia. The presence of an Ordovician volcanism associated with local listric faults is not an argument to support the record of the Sardic Phase".

*B.2) In the oriental areas included in the paper, a short comment is made on the presence of basic volcanic bodies at the considered ages, which are not included in the research. In order to have a more consistent geological setting between the several domains, the Truchas (Truchas syncline) and Santo Adrião (east Morais) basic volcanics (MAGNA map, Dias da Silva et al., 2011; Dias da Silva et al., 2016) could also be included as a minor comment*

This has considered in the above paragraph (see previous remark).

*B.3) As figure 2 is an important asset of the paper, especially for readers with scant knowledge of some areas geology, several improvements would be inserted. In the CIZ the Cambrian sequence reflects only the southern part; lacking the Terreneuvian and Series 2 sequence of the northern part (Díez Balda, 1986). In the Upper Ordovician, the unconformity may well be reflected. A minor mistake exists in La Aquiana limestone's spelling. In Truchas area, the Upper Ordovician is mainly terrigenous and divided in Casaio, Rozadais and Losadilla formations,*

*being the limestone (La Aquiana) a minor and discontinuous unit; for this reason the now reflected sequence is misleading. In the GTMZ only the Upper Parautochthon must be included, as the Lower Parautochthon is a sequence of Variscan synorogenic origin displaying ages from Uppermost Devonian to Carboniferous (Oliveira et al., 2019). In the UP the sequence does not reflects the proposed for Dias da Silva et al. (2016), as it doesn't includes the voluminous Middle-Upper Ordovician acid and basic volcanism of the Morais Complex (Peso). The labels and legend must be revised as some mistakes have been found (including lost references which will help to fit data and sketch, especially on the older known ages for the Ollo de Sapo).*

The Ediacaran-Cambrian lithostratigraphic chart of what you call "northern part of CIZ" is in need of re-evaluation. Díez Balda (1986) proposed a lithological subdivision and Valladares et al. (2000, 2006) proposed a subdivision into labelled facies associations (Ediacaran units I to IV, and Cambrian Units V to XII underlying the Tamames Sandstones). In our opinion, adding such a nomenclature would complicate the readiness of the figure and we prefer avoiding it because we have not used these units in the paper. The remaining proposals have been updated. Thanks.

*Regarding the suggestions marked in the pdf file:*
*Lines 74: As no new data supporting the relationship between magmatism and the Sardic unconformity are offered, it will be better to state both events are coeval.*

The age of the Sardic volcanism ranges from Early to Late Ordovician, and the gaps associated with the Sardic unconformity (uplift + erosion + marine transgression and deposition) are about 16-20 m. In short, the gap is 16-20 m.y., whereas the magmatic activity took place during a time span of about 25-30 m.y. (from 475 to 445 Ma). Broadly speaking, both ranges can be considered as "contemporaneous".

*Line 77: A comment on the presence or absence of folding and schistosity related to the Sardic Phase must be added.*

t is done.

Line 126. "A SW-NE palaeogeographic transect" is in today coordinates. We disagree, today we would follow a W-E trend, and all the palaeogeographic reconstructions of West Gondwana, during Cambrian times, show a SW-NE margin of Gondwana, e.g.:

[Figure]

Line 278. We have delated "Furongian strata" as it misled the real information. Thanks.

Figures: they are updated following your remarks, except the Ediacaran-Cambrian lithostratigraphic subdivision in Salamanca, which is in need or re-evaluation.

Thanks again for your editing revision.

---

## Editor Decision (ED2)

Dear Authors,

After having examined the revised manuscript and your response, I have noticed that some important issues have not been corrected and need further consideration before the acceptance of the manuscript. I will explain below the points:

**1. Classification of Variscan Zones into autochthon or parautochthon - allochthon**

In your response you said:

*"Taking into account that the topic editor proposed us to follow one of two options (either subdividing the "autochthon" or the "allochthon-parautochthon"), we have added a subdivision of the "allochthons" in figure 1A. The topic editor did not indicate which "allochthon-parauthochthon" had to be subdivided from those included in the figure (i.e., the Galicia-Trás-os-Montes Zone, the northern French Massif Central and the northern edges of Sardinia and Corsica) so we have subdivided all of them. The final result is, in our opinion, hardly readable. The reason why we added figures 1B-to-1D was to help any potential reader to find the samples reported in the paper, but the addition of more information in figure A gives rise to an extremely complex figure that is not necessarily useful."*

It seems to be a misinterpretation here. In my last and previous comments I clearly explained the need to include the variscan zones (all of them) either in "autochthon" or the "allochthon-parautochthon". In no case has it been requested a subdivision of the "allochthon-parautochthon", but a proper classification of the zones (all of them). In no case was the identification of the complexes, individual units or sub-domains requested, as in a confusing way it appears now it the new version. Considering the scale of the map, this would be a nonsense

Besides, in the last uploaded version, in the Fig. 1A, there are none of the requested changes. The authors maintain their categories: "Variscan autochthons-parautochthons" and "Variscan-allochthons" without any single explanation about the grouping criteria. In the caption it is said this figure is a modification from Pouclet et al. (2017), which indeed is a modification of Ballevre et al. (2009). There are new and updated representations of such a reconstruction (e.g. Ballevre et al. 2014). Some of them come from the authors of this paper, like the Fig 1 of Casas, J.M., Murphy, J.B. 2018. *Unfolding the arc: the use of pre-orogenic constraints to assess the evolution of the Variscan belt in Western Europe. Tectonophysics 736, 47–61*, where variscan zones' grouping criteria into parautochthon, autochthon and allochthon categories, are clearly defined and show scientific consistency.

I have reviewed again the manuscript text and there are not a single data, at its present form, which support a change in the scheme of variscan zones proposed by Ballevre et al. (2014), Casas and Murphy (2018), Martínez Catalán et al. (2019), Azor et al. (2019)... to name a few ones, so It has to be assumed the authors does not have any scientific evidence to support his new classification. As a consequence please modify the Fig 1A as requested in the previous revision.

Is nevertheless surprising that, instead of making the required changes, the authors have chosen to make a free and imaginative interpretation, adding more errors and confusion to the old figure 1A. I cannot see the purpose of incorporating in the Fig 1A, the name of, for example, the allochthonous complexes of the NW of Iberia, but at the same time leaving as part of the allochthonous, the schistose domain (parautochthon). In parallel, I cannot find any argument in the text supporting the inclusion of e.g. Essarts, Ile-de-Groix or Bois-de-Céné units (Armorican Massif) as part of the autochthon, having been classified as allochthon long time ago (e.g.

Ballevre et al. 2014 and references therein). Similar errors are found in the French Central Massif (e.g. Parautochthon Micaschits, etc).

What is the intention of the new addition in the Iberian Allochthon, the Verín-Bragança synform? Should we consider it a new allochthonous complex? Seeing those kind of things, I suspect that not all the co-authors have had access to this version of the article, because it represent a complete nonsense, equivalent to states that the Main Himalayan Thrust is a Neoproterozoic structure.

The authors have to have clear that the problem of the Fig 1A is not, *the addition of more information,* but the number of errors it has and the absence of arguments supporting their new ideas.

**2. Cambro-ordovician Lithostratigraphy**

**A)** *"The second question is focused on the (litho-)stratigraphic units previously published exclusively in the Salamanca area of the Central Iberian Zone. As already explained in a former response, we cannot understand why the topic editor requests us to cite some previous works related only to his area of work and ignores other similar reports proposed for other areas included in our work.*

It seems obvious that a request has been made to correct what is lacking. What is correct, or what the reviewers have already pointed out, is not included in my report.

*What are about other stratigraphic units that, like the Salamanca area, which are not reported in this paper because we selected no samples from there? No interest about them...*"

Again the problem here is not a personal interest. The authors are trying to make large-scale correlation based on previous and new datasets. However, the criteria for using some data (particularly stratigraphic) and not others are not clear. In the answer now it seems that they are only the data of the samples directly obtained for the study. This is not strictly true since they use previous results and data from the literature. In this line, for example, the information published about the Salamanca sector was not selected, but at no point in the text do the authors indicate the criteria they followed in the selection. In the replies the authors give some reasons but never include them in the manuscript.

In connection with this issue, it is surprising what the authors say in their response:

*"This paper offers no lithostratigraphic revision of the Central Iberian Zone; this should be obvious for any reader"*

And in the caption of the Fig 2 authors said: *"Stratigraphic comparison of the Cambro-Ordovician successions from the Central Iberian Zone [...]".*

The text and the figure 2 invite the reader to find a correlation of the cambro-ordovician successions across the orogenic zones as a whole (CIZ, GTMZ...). But what he finds is a correlation of certain parts of the zones, taken as a whole. I have no problem with showing a correlation between the specifically sampled zones, but it must be clearly explained in the figure and/or its caption. This can be done by indicating which parts of the existing information in an area have been omitted and why (my proposal), or by indicating that the correlation is made only among the areas sampled in this work, and therefore does not represent a correlation with all the existing data.

Besides, the comment introduced in the Fig 2 caption: "*the northern Central Iberian Zone, in the vicinity of Salamanca, is not included here (Díez Balda et al., 1990)*" is not correct, because the Diez Balda subdivision applied not only in the Salamanca city, but to the province.

**B)** *"However, we disagree in citing Valladares et al.'s works. Our position is not related to any animosity against the author, Prof. Isabel Valladares, whom we know well. Our refusal is based on a conceptual misleading that was explained in a previous response to the topic editor, but we have no troubles to repeat it.*

*In her PhD thesis and following papers, Isabel Valladares defined FACIES ASSOCIATIONS (labelled I to XV), not FORMATIONS, and she suggested some broad correlations with previous stratigraphic units. As you know (or you should), a facies association (characterized by an assemblage of facies and useful for environmental interpretations) and a lithostratigraphic unit (characterized by its lithology and useful for mapping) are two completely different things. Therefore, we disagree with your insistence on citing works that describe facies associations as if they were lithostratigraphic formations and thus hampered any stratigraphic correlation. This sounds a serious misconception of geological concepts. "*

I sincerely appreciate the explanation about facies and lithostratigraphic units, but again, the problem here is not a "conceptual misleading", but the absence of clear criteria to include some data and refuse others. It seems like the reader has to blindly believe on authors' personal selection. In the response is clear that the authors have a reason for not to use Valladares' works, but refuse to explain them in the main text. Beside, it is important to note that the authors in the Figure 2 caption said:

*"Stratigraphic comparison of the Cambro-Ordovician successions [...]"*

It might be obvious for the authors that "Stratigraphic... successions" means "only lithostratigraphic units", but, should not be more precise to say "Comparison of the Cambro-Ordovician lithostratigraphy...? It would suddenly clear up the confusion that for an interested reader of a wide-audience journal like Solid Earth could generate.

At this point the affair is still up:

1. - The authors have to complete the required changes in Figure 1A and make sure to include all the variscan zones either in the autochthon or in the allochthon-parautochthon. I suggest following one of the co-authors reconstruction (Fig. 1 Casas & Murphy, 2018) or, e.g. Martínez Catalán et al (2019), see references at the end of this letter.

2. - Please remove the names added in the last version of the Fig. 1A: OC, MC. COC, BZ, Mv, Gu, Br, Ma, Mo, VC, IZ and particularly the "psychedelic" Verín-Bragança syncline (VBS) allochthon.

3. Consider to change in Fig 2 caption "Stratigraphic... successions" by formal stratigraphic definitions like "Cambro-Ordovician lithostratigraphy". In the same caption change " *the northern Central Iberian Zone, in the vicinity of Salamanca..." by "the northern Central Iberian Zone, in Salamanca..."*

4. Clarify whether in the main text or in the caption of Fig 2 the criteria for stratigraphic data selection. This might be something like "lithostratigraphic data used in the correlation was restricted to the sampled areas".

I am therefore returning the manuscript with a request of minor revision.

Sincerely,

Juan Gómez Barreiro

References:

Ballèvre, M., Martínez Catalán, J.R., López Carmona, A., Abati, J., Díez Fernández, R., Ducassou, C., Pitra, P., Arenas, R., Bosse, V., Castiñeiras, P., Fernández-Suárez, P., Gómez Barreiro, J., Paquette, J.L., Peucat, J.J., Poujol, M., Ruffet, G., Sánchez Martínez, S., 2014. Correlation of the nappe stack in the Ibero-Armorican Arc across the bay of Biscay: a joint French–Spanish Project. In: Schulmann, K., Martínez Catalán, J.R., Lardeaux, J.M., Oggiano, G. (Eds.), The Variscan Orogeny: Extent, Timescale and the Formation of the European Crust. Geological Society, London, Special Publications, vol. 405. pp. 77–113.

Martínez-Catalán, J.R., 2011. Are the oroclines of the Variscan belt related to late Variscan strike-slip tectonics? Terra Nova 23, 241–247. http://dx.doi.org/10.1111/j. 1365-3121.2011.01005.x.

Azor, A., da Silva, Í.D., Barreiro, J.G., González-Clavijo, E., Catalán, J.M., Simancas, J.F., Poyatos, D.M., Pérez-Cáceres, I., Lodeiro, F.G., Expósito, I. and Casas, J.M., 2019. Deformation and Structure. In The geology of Iberia: A geodynamic approach (pp. 307-348). Springer, Cham.

Martínez Catalán, JR., Collett, S., Schulmann, K., Aleksandrowski, P., and Mazur S., 2019 "Correlation of allochthonous terranes and major tectonostratigraphic domains between NW Iberia and the Bohemian Massif, European Variscan belt." International Journal of Earth Sciences: 1-27.

---

## Author Response (AR3)

Dear editor,

Thanks for your comments and suggestions proposed to "ensure quality and clarity of the submitted work". We have uploaded a new pdf file with only one modification in Figure 1A.

**Point 1**. You request some explanations in the text to justify why we consider "CZ, WALZ, CIZ as a par-autochthon" and you offer us a selection of data for such a discussion.

The purpose of figure 1 is to help any potential reader to place the rock samples in a tectonostratigraphic context. Figure 1A offers a tentative Variscan reconstruction where the tectonostratigraphic units are subdivided according to only two (coloured) features: Variscan allochthons and non-allochthons. Our selection cannot follow all the information available in the literature because there are as many Variscan models as authors. You claim that CL, WALZ and CIZ are Variscan autochthons because this is a kind of paradigm, but what's about the remaining units coloured in grey? Should we also differentiate them according to other Variscan paradigms? We assume that, for you, OMZ is correctly labelled as autochthon or parautochthon, but this is against any paper co-authorized by Ricardo Arenas where OMZ is considered as a Variscan allochthon. CLZ, WALZ and CIZ are different parts of the foreland and fold belt of a Variscan Orogen that you consider "Autochthon". We understand that this foreland and fold belt could be considered as a relative autochthon by comparison with the NW Iberia allochthon complexes, but not as a Variscan autochthon by its own structural framework.

You say nothing about the OMZ, the Pyrenees, the Occitan Domain and SW Sardinia. Are there also autochthons for you? That's why we tried to summarize all these domains as (par-)autochthonous units by comparison with the distinct Variscan allochthons of SW Europe. In any case, this cannot be summarized with a sentence in our text, as you suggest, but it is in need of a deep discussion that would need a complete paper or a chapter in a book. You did not like our solutions, and you want to see the word "autochthon" here because this is your paradigm. No problem. We have labelled all these units of SW Europe "autochthons and parautochthons" as a solution to distinguish them from the remaining "allochthons". Hope this will solve your query. Ricardo Arenas will surely disagree but he has neither edited nor reviewed our paper, and we hope he will understand why we have selected such a solution for our figure 1A.

**Point 2**. You consider that adding Díez Balda (1986) and Valladares et al. (2000, 2006) lithostratigraphic nomenclatures for the Cambrian of "the northern part of the CIZ" "would help and guide the interested reader to understand the state of the art". First at all, Díez Balda and Valladares et al.'s nomenclatures are completely different and uncorrelatable. This is clearly stated in Valladares' works, where the contacts of the Monterrubio and Aldeatejada formations outside their type area are not identified. Other problems for Díez Balda's nomenclature is the selection of a single toponymical term to name two formations (Tamames Limestone and Tamames Sandstone formations), which fails the stratigraphic rules of nomenclature. In addition, as clearly stated by the own authors, Valladares terms (I to XII) are facies associations and not lithostratigraphic units. If the authors were not able to correlate these units outside the Ciudad Rodrigo-Hurdes-Sierra de Gata domain, why should we solve this stratigraphic puzzle in our paper considering that these units are not used in our text? A Portuguese reviewer could request a justification for the absence of the Portuguese lithostratigraphic nomenclature from the Central-Iberian Zone. If we add the terminology of the Ciudad Rodrigo-Hurdes-Sierra de Gata domain and the Portuguese areas, the resulting figure would be incomprehensible and full of correlating hypotheses that would need explanations in the text. Figure 2 is there to illustrate the stratigraphic (mainly chronostratigraphic) setting of the analysed rock samples, trying to be as simple and comprehensible as possible. We are not trying to offer the state of the Cambro-Ordovician stratigraphic art in CIZ for the potential readers, but simply trying to place the study rocks in a stratigraphic context as simple and comprehensible as possible.

In short, we could blindly follow your indications to see our paper finally accepted and focus our attention on other questions. This would be the easiest solution, but we do not understand why you are suggesting such questions that, in our opinion, are not "ensuring quality and clarity". We kindly disagree with your second revision (not edition) and have explained above why. Thanks again.

---

## Editor Decision (ED3)

Dear authors,

Thanks for your kind reply and effort to improve the quality of your manuscript. Some issues are now clearly solved with the incorporation in the manuscript of precise information. Now every reader can get a clear idea of the meaning and regional scope of your Cambro-Ordovician lithostratigraphic compilation (Figure 2). Well, we have taken a step forward.

However, I still find Figure 1A inaccurate and misleading. I have carefully read the new publication included, Alvaro et al. 2020, hoping that it would shed some light on the new interpretation that the authors make of the variscan zones. Unfortunately there is no data to support that, for example, the Schistose Domain (GTMZ) should be considered part of the allochthonous, or the imaginative classification of the CIZ, WALZ or CIZ as "autochthonous - parautochthonous" category.

I am sure that I am not the most prepared reader, but I strive to understand these surprising shifts in the interpretation of geological evidence presented in recent decades, such as the authors' proposal to consider the Ile-de-Groix, Essarts or Bois-de-Céné units as part of the Variscan autochthon, to name a few issues.

The problem here is not the Pre-Variscan interpretation of the variscan zones, but the Variscan one which is used in Fig.1A (autochthon, parautochthon, allochthon).

Some replies to your doubts:

 1. *Why should we include those Variscan zones that are neither described nor considered in our paper? I guess you refer to the non-periGondwanan South Portuguese Zone or terrane which was far from the southwestern*

My request was literally:

[...] the need to include the variscan zones (all of them) either in "autochthon" or the "allochthon-parautochthon".

The sentence means: to *include* (put in as part of a group) all the variscan zones depicted in Fig. 1A. Not to include all the Variscan zones! If the SPZ was not mentioned in the original figure and text, there is no need to include it in this version, unless the authors consider it of interest to complete the interpretative framework.

 2. *We have adapted the figure caption of Figure 1A as follows: "Pre-Variscan reconstruction of the Variscan tectonostratigraphic units bearing Cambro-Ordovician exposures REPORTED IN THIS WORK, from the south-western European margin of Gondwana; based on Pouclet et al. (2017) and Álvaro et al. (2020)."*

As explained before, the new publication (Alvaro et al 2020, GSL) uses a figure/classification with similar errors to the present manuscript, without clearly explaining the criteria applied in the new interpretations, not of Pre-Variscan

context, but of the classification of the Variscan zones into autochthon, parautochthon or allochthon. So, unfortunately is not a solution of the issues presented in my previous review.

As stated in previous reviews, In order to correct the Fig 1A, please follow your own work like the Fig 1 of Casas, J.M., Murphy, J.B. 2018. *Unfolding the arc: the use of pre-orogenic constraints to assess the evolution of the Variscan belt in Western Europe. Tectonophysics 736, 47–61*, where variscan zones' grouping criteria into parautochthon, autochthon and allochthon categories, are clearly defined and show scientific consistency.

In order to solve those issues and make this manuscript suitable for publication in SE:

a)- The authors have to complete the required changes in Figure 1A and make sure to include all the variscan zones either in the "autochthon" or in the "allochthon-parautochthon" categories. I suggest following one of the co-authors reconstruction and criteria (Fig. 1 Casas & Murphy, 2018) or, e.g. Martínez Catalán et al (2019), see references in my previous letter. (in my previous reviews there are more details)

b)- Please remove a small line emerging from Cabo Ortegal complex in Fig. 1A.

c)- If the SPZ is not used in your discussion please remove it from Fig.1A. Alternatively, if you consider it provides a better picture of the context, classify (= colour it) according to the same criteria than explained in a).

I am therefore returning the manuscript with a request of minor revision.

Sincerely,

Juan Gómez Barreiro

---

## Author Response (AR4)

Dear topic and chief-executive editors,

Thanks for your messages. We have just uploaded the last version of our paper ref. se-2020-45.

According to the last revision (not edition) of the topic editor, there were two aspects that had to (not should) be improved.

1. A "new version with the corrections in Figure 1A including the different zones either in the autochthon or in the allochhon-parautochthon".
2. "Besides, a good compromise about Ediacaran-Cambrian lithostratigraphic chart issue is to declare well into the main text or the caption that you are aware of other evidence (e.g. Diaz Balda, Valladares et al...) but you are taken, as a reference for the whole ZCI, the stratigraphic model developed in the southern part of the ZCI, because of its consistency/coherency/ degree of development".

The chief-executive editor considered that she "cannot see any bias occurring by the minor changes recommended, which suggest already a compromise between different possibilities. Thus, my final decision is that the TEs comments are not disturbing your manuscript flow, so that you should consider them before publishing is possible".

For us, it is a surprise to read these remarks but we have decided to follow your queries and opinions in order to finish with this affair, despite our complete disagreement with the management of this manuscript. If you expend a couple of minutes reading our explanations, you would understand why.

1. Taking into account that the topic editor proposed us to follow one of two options (either subdividing the "autochthon" or the "allochthon-parautochthon"), we have added a subdivision of the "allochthons" in figure 1A. The topic editor did not indicate which "allochthon-parauthochthon" had to be subdivided from those included in the figure (i.e., the Galicia-Trás-os-Montes Zone, the northern French Massif Central and the northern edges of Sardinia and Corsica) so we have subdivided all of them. The final result is, in our opinion, hardly readable. The reason why we added figures 1B-to-1D was to help any potential reader to find the samples reported in the paper, but the addition of more information in figure A gives rise to an extremely complex figure that is not necessarily useful.

2. The second question is focused on the (litho-)stratigraphic units previously published exclusively in the Salamanca area of the Central Iberian Zone. As already explained in a former response, we cannot understand why the topic editor requests us to cite some previous works related only to his area of work and ignores other similar reports proposed for other areas included in our work. His view of this issue looks biased and this partial vision should be heralded by no editor. What are about other stratigraphic units that, like the Salamanca area, which are not reported in this paper because we selected no samples from there? No interest about them.

As above, and in order to finish with this discussion, we have cited Díez Balda et al. (1990) as the origin of another lithostratigraphic subdivision of the Salamanca area, and we hope that the topic editor will be satisfied with the special treatment we are finally offering exclusively to the Salamanca area, though it should be noted that there are NO analyzed samples from there in our paper. This paper offers no lithostratigraphic revision of the Central Iberian Zone; this should be obvious for any reader.

However, we disagree in citing Valladares et al.'s works. Our position is not related to any animosity against the author, Prof. Isabel Valladares, whom we know well. Our refusal is based on a conceptual misleading that was explained in a previous response to the topic editor, but we have no troubles to repeat it.

In her PhD thesis and following papers, Isabel Valladares defined FACIES ASSOCIATIONS (labelled I to XV), not FORMATIONS, and she suggested some broad correlations with previous stratigraphic units. As you know (or you should), a facies association (characterized by an assemblage of facies and useful for environmental interpretations) and a lithostratigraphic unit (characterized by its lithology and useful for mapping) are two completely different things. Therefore, we disagree with your insistence on citing works that describe facies associations as if they were lithostratigraphic formations and thus hampered any stratigraphic correlation. This sounds a serious misconception of geological concepts.

We have uploaded the revised version including these two issues and hope that the paper will be definitively accepted in its present form.

Kind regards

JAVIER ALVARO

---

## Editor Decision (ED4)

Thank you for your kind and passionate response. I fully agree that the dialectical confrontation of ideas is the basis for science to advance and improve. The process may seem long, as when we learn to read, but the result is certainly worthwhile. Having said this and although the digressions raised by the author are certainly interesting, I think it is important that authors focus their efforts on presenting scientific arguments.

Although reasonable corrections have been made to the text and figures of the article, the answers have never been directed at answering the only question that remained: Why are the Variscan zones classified in the new way in which they appear in Figure 1A? This is a relevant aspect in the context of the special volume that concerns us.

The repeated lack of data in the responses on this point, the surprising use of ad hominem-type argument fallacies in a scientific context, as well as the less elegant resort of threatening the editorial team with withdrawal of the article, clearly indicate that the author's fundamental argument for supporting the model in Figure 1A is "just because".

It is great the author decided to show different reconstructions of the variscan belt in his last letter. This is a good starting point. But please, note that in addition to beautiful figures, there is a clear explanation of the scientific data, and arguments that support those figures, in the text of those articles. This is the only request throughout this editing process that the author seems unwilling to share with everyone. I sincerely hope that  the author's recent readings will lead him to explain this in future publications. I have also noticed that some recent references are lacking like this one from 2019: https://doi.org/10.1007/s00531-019-01800-z

[Figure]

In this line, a published inconsistency is an inconsistency (or error). In Fig1A in Alvaro et al 2020 GSL there are similar inconsistencies in their use of variscan

parautochthon, allochthon and autochthon, particularly when they are compared to their main text.

It is much appreciated that the author has specified the extension of the application of his lithostratigraphic data of the Cambrian and Ordovician, initially presented as an over-generalization of the whole of the central Iberian zone, as there were notable absences not initially declared in the manuscript. Similarly, in this last version, he has abandoned the idea of including the Portuguese Southern Zone in the debate, something that was not requested in the revisions, or defining new allochthonous complexes such as that of Verín-Bragança. I am sure that the latter was due to a totally involuntary drawing flaw on the part of the author. As probably is the inclusion of the Schistose Domain among the allochthonous sequences, or perhaps it is simply an incipient color blindness that surreptitiously affects my perception of the chromatic shades of figure 1A.

Fortunately after the last corrections the bulk of the article is not affected by the problem detected in figure 1A and presents excellent data (e.g. geochemistry) that will be very useful to understand the evolution of the Variscan Orogen in a broad sense. I hope that the interested reader will appreciate this balance between what can reasonably be expected from an editing process, and the argumentative licenses that can inevitably lead us all to force arguments and manners because of the passion with which we live this science.

Although it would have been preferable to see the argumentation reflected in the manuscript itself, I refer the reader to the fruitful discussion held throughout the process, to form his/her own idea about the interpretation of the different zones of the Variscan Orogen.

I thank the authors again for their effort and patience.

Cordially yours

Juan Gómez Barreiro

---

## Author Response (AR5)

Dear topic editor and third reviewer,

Regarding your last queries:

1. You want to see "all the Variscan zones either in the autochthon or in the allochthon-parautochthon in Figure 1A". It is clear that you cannot see a paper focused on pre-Variscan issues without finding the classical Variscan subdivision of the Iberian Massif. But this paper is not focused on ALL the Variscan zones of the Iberian Massif. Why should we include those Variscan zones that are neither described nor considered in our paper? I guess you refer to the non-periGondwanan South Portuguese Zone or terrane which was far from the southwestern European margin of Gondwana during Cambro-Ordovician times. If this was not a peri-Gondwanan sector during Cambro-Ordovician times neighbouring the south-western European Gondwana margin (what was clearly stated in the figure caption) why should we highlight it? Simply, because you want to impose us the models are in your mind considering that other models are not scientific.

We have adapted the figure caption of Figure 1A as follows: "Pre-Variscan reconstruction of the Variscan tectonostratigraphic units bearing Cambro-Ordovician exposures REPORTED IN THIS WORK, from the south-western European margin of Gondwana; based on Pouclet et al. (2017) and Álvaro et al. (2020)." In addition, we have added the setting of the South Portuguese Zone (not coloured it) though it is not reported in our work, it has no Cambro-Ordovician exposures and was not fringing the targeted Gondwana margin during Early Palaeozoic times. Of course, this figure is based on Pouclet et al. (2017) and Álvaro et al. (2020 and already online); we are sure you consider these works as scientifically unacceptable because you only understand your own dogmas, but if you try to open a little your mind you would be able to remember the times when you were still able to accept other ideas than yours.

2. In your previous revision, you "suggested" (or wanted to impose) "a good compromise about Ediacaran-Cambrian lithostratigraphic chart issue is to declare well into the main text or the caption that you are aware of other evidence (e.g. Diaz Balda, Valladares et al...) but you are taken, as a reference for the whole ZCI, the stratigraphic model developed in the southern part of the ZCI, because of its consistency/coherency/ degree of development".

This was done in our last version, where we reported Díaz Balda et al. (1990) and explained that we had selected no samples from the Salamanca area, another of your obsessive ideas. We explained you the difference between lithostratigraphic units and facies associations in order to avoid citing papers that include no lithostratigraphic subdivisions (cf. Valladares' papers). Now you write the sentences you want to read in our paper behaving like a co-author. As explained several revisions ago, there are other areas with their own lithostratigraphic subdivision that are not included in our figure 2, and not only Salamanca, Salamanca and Salamanca. Looking for consistency, we have included a selection of them in the new version, because Salamanca is not special for us, only for you, which says: "Cambro-Ordovician lithostratigrahic chart OF THE AREAS STUDIED IN THIS WORK from the Central Iberian Zone, Galicia Trás-os-Montes Zone, Occitan Domain, Eastern Pyrenees and Sardinia; modified from Álvaro et al. (2014b, 2016, 2018), Pouclet et al. (2017) and Sánchez-García et al. (2019); other areas, such as the Ciudad Rodrigo-Hurdes-Sierra de Gata Domain (Díez Balda et al., 1990) and the Portuguese sector (Medina et al., 1998; Meireles et al., 2013) in the Central Iberian Zone, the central Pyrenees (Zwart, 1979; Laumonier et al., 1996), the Albigeois Mountains from the Occitan Domain (Guérangé-Lozes and Alsac, 1986; Pouclet et al., 2017) and northern Sardinia (Elter et al., 1986), are not included here". We think this is not too complicate to understand. There is even a section entitled "Material and methods" were the study areas are clearly documented.

We are looking forward to receiving your new queries.

Kind regards

JAVIER ALVARO on behalf of ALL the authors

---

## Author Response (AR6)

Dear topic and executive editors,

This is our last letter addressed to you.

We have just uploaded the ninth revised version of our manuscript. As suggested by the third reviewer/topic editor, we have removed SPZ and an emergent line in our figure 1A. We hope that, with these last changes, the abnormally boring and tedious reviewing process of this paper will be finished. Usually, reviewing a contribution is a challenging and exciting process leading to the improvement of the first submitted version. Unfortunately, this is not the case now and, in our opinion (three words that the topic editor never uses because he has not opinion, he simply knows), this is mainly due to an important misunderstanding of editor's role. The editor has to assure that the contribution is aligned with the quality standards of the journal and the topic of, in this case, special issue. The role of the editor is not to push dogmatically the authors to write what he wants to read because he knows the truth. Discrepancy is the basis of scientific progress in any matter, including geology.

The last problem of this paper still deals with figure 1A, where the topic editor wants to see the Variscan subdivision he pleases, and no other. Otherwise he will never accept our paper. The rest of the paper is out of topic editor's mind. This figure is based on two papers, the last one co-authorized by some of us in 2020. In his last letter, the statement of the topic editor is clear about Álvaro et al. (2020): "this publication uses a figure/classification with similar ERRORS to the present Ms (…) so it is not a solution of the issues presented in my previous review". So the topic editor is deciding here which papers publish errors in their texts and figures and which papers not. Sincerely, this is a presumptuous and arrogant sentence that labels topic editor's nature.

There are many Variscan reconstructions of southwestern Europe, each one highlighting different complexities. According to the topic editor, there is only one correct and the remaining published Variscan subdivisions are full of errors. Below you will find four selected subdivisions of the Variscan massifs in SW Europe, each one with different alternatives and contradicting interpretations.

MODEL A. This model has been published by Álvaro et al. (2020) in the Geological Society of London with an editor, two reviewers and three authors (two of them have written this letter). As stated above, the topic editor has decided that this figure includes "errors". So, the three authors, the two reviewers and the editor were full of flawed ideas that only our topic editor is able to see.

MODEL B. This is the good model, because the topic editor co-authorized it (Martínz Catalán et al., 2019). But there are several "hard interpretations" here (we do not say "errors" because we are polite, Dr. Barreiro). Topic editor wants to see autochthons in our paper, but no autochthons are distinguished in this model, only parauthochtons and allochthons (e.g., GTMZ, SMD and northern Massif Central). Some curious oroclines occur as well, such as the Cantabrian Orocline together with a partial Central Iberian Orocline. The subdivision of "zones with Cadomian imprint" and "with Early Ordovician magmatism" is wholly arbitrary as this reconstruction ignores, at least, the Cadomian imprint in the Central Iberian and Cantabrian Zones and in the Iberian Chains.

MODEL C. According to this model, the allochthons are located at the GTMZ (the whole zone, not parts of it), the SPZ, the SAD (not the central/northern domains) and the northern Massif Central; no parautochthons are highlighted here; the Central Iberian orocline shows a different outline than the previous version; after Shaw & Johnston (2016). Of course, these authors would have never published this paper with Dr. Barreiro as topic editor. They would be still modifying their figure 1.

MODEL D. Another different model where the Ossa-Morena Zone is a lateral prolongation of the GTMZ allochthon (it is described as an allochthon contradicting editor's dogmatic model), the SAD and NAD are allochthons but the CAD is an autochthon (new differences); the Iberian oroclines are the Central Iberian and the Ibero-Armorican (not the Cantabrian) Oroclines; taken from Arenas et al. (2016, 2019) and Díez Fernández et al. (2020).One of us (JJA) has discussed this model with Ricardo Arenas several times; we disagree but we gently disagree, avoiding situations such as that forced by our topic editor. Of course, these papers would have never been published with Dr. Barreiro as topic editor.

We could add further models and paradigms different from those dogmas blindly defended by the topic editor but we are sure he will never been persuaded by us. For him, this is a personal (not a scientific) matter.

We consider no further changes are necessary and we will add no further modifications. In case the paper is not accepted in its present form, we will submit it to another journal. This is not our preferred scenario because, for multiple personal and professional reasons, we would like to offer homage to José Ramón Martínez Catalán, an OPEN-MINDED person who RESPECTS other ideas. Whatever the case, we are sure that, when explained, José Ramón will understand our point of view and our homage in another journal.

Looking forward to hearing you soon,

JAVIER ALVARO

[Figure]

Álvaro et al. (2020)

Martínez Catalán et al. (2019)

[Figure]

Shaw & Johnston (2016)

[Figure]

Arenas et al. (2016, 2019) and Díez Fernández et al. (2020)